# Structural insights into Wnt/β-catenin signaling regulation by LGR4, R-spondin, and ZNRF3

Yuxuan Peng[1,3], Akiko Fujimura[1,3], Jinta Asami[1], Zhikuan Zhang[1], Toshiyuki Shimizu [1] ✉ & Umeharu Ohto [1,2] ✉

Leucine-rich repeat-containing G protein-coupled receptor 4 (LGR4) plays a critical role in regulating the wingless-related integration site (Wnt) signaling pathway and is essential for organ development and carcinogenesis. LGR4, along with its ligand R-spondin (RSPO), potentiates Wnt/β-catenin signaling by recruiting its signaling suppressor, E3 ligase Zinc and Ring Finger 3 (ZNRF3), and inducing its membrane clearance. However, detailed mechanisms underlying this process remain unknown. In this study, we present the cryo-electron microscopy structures of human LGR4, the LGR4-RSPO2 and LGR4-RSPO2-ZNRF3 complexes. Upon RSPO2 binding, LGR4 undergoes no significant conformational changes in its transmembrane and extracellular domain structures or their relative orientations. LGR4, RSPO2, and ZNRF3 assemble into a 2:2:2 complex with the ZNRF3 dimer enclosed at the center. This ternary arrangement and forced dimerization of ZNRF3 likely underpin how LGR4 and RSPO2 potentiate Wnt/β-catenin signaling by sequestering ZNRF3 from Wnt receptors and facilitating its auto-inactivation. This study provides a structural basis for understanding the regulatory mechanism of Wnt/β-catenin signaling through the LGR4-RSPO2-ZNRF3 pathway and may offer opportunities for future drug development targeting this axis.

The Wnt/β-catenin signaling pathway is a fundamental mechanism that regulates essential cellular processes, such as proliferation, differentiation, migration, and tumorigenesis[1–3]. Dysregulation of Wnt/β-catenin signaling is associated with various diseases, including impaired tissue repair, skeletal disorders, neurodegenerative conditions, and multiple types of cancers[4,5]. Extracellular Wnt proteins bind to the seven-transmembrane receptors Frizzleds (FZDs) on the cell surface, in conjunction with single-transmembrane low-density lipoprotein receptor-related protein (LRP) 5/6, to initiate the Wnt/β-catenin pathway. This interaction promotes the accumulation of cytosolic β-catenin, which then translocates to the nucleus and activates TCF/LEF transcription factors, triggering the expression of target genes[6,7].

Wnt/β-catenin signaling pathway activation is regulated by a set of auxiliary proteins, including Zinc and Ring Finger 3 (ZNRF3), Ring Finger Protein 43 (RNF43), Leucine-rich repeat-containing G-protein coupled receptor 4/5 (LGR4/5), and R-spondin (RSPO)[8–11] (Supplementary Figs. 1–4). Single-transmembrane E3 ubiquitin ligases ZNRF3/RNF43 downregulate Wnt/β-catenin signaling by mediating the ubiquitination and degradation of a subset of FZDs and LRP5/6[12,13]. Cell surface-expressed LGR4/5, members of the class A G-protein coupled receptors (GPCR) family, have an extracellular domain (ECD) comprising 17 leucine-rich repeats (LRRs), followed by a seven-transmembrane domain (TMD), and an intracellular domain (ICD) (Fig. 1a)[14]. LGR4/5 amplifies Wnt/β-catenin signaling in a manner dependent on their interaction with the RSPO family of ligands[9]. Unlike the glycoprotein hormone receptors (LGR1-3)[15–17], this function of LGR4/5 is G protein-independent[9,11]. ZNRF3/RNF43, LGR4, and RSPO form a ternary complex on the cell surface, which relieves ZNRF3/

[1]Graduate School of Pharmaceutical Sciences, The University of Tokyo, Tokyo, Japan. [2]Graduate School of Frontier Sciences, The University of Tokyo, Kashiwa, Chiba, Japan. [3]These authors contributed equally: Yuxuan Peng, Akiko Fujimura. ✉e-mail: shimizu@mol.f.u-tokyo.ac.jp; umeji@g.ecc.u-tokyo.ac.jp

RNF43-mediated suppression of Wnt/β-catenin signaling and thereby potentiates pathway activation[11,18,19]. This may occur by facilitating the membrane clearance of ZNRF3/RNF43 through auto-ubiquitination[19,20]. Previous crystallographic studies have provided substantial structural information on this axis[14,21–25]. However, this information is limited to the ECD of these proteins. Consequently, a comprehensive understanding of the mechanism of Wnt/β-catenin signaling regulation has been hindered.

In this study, using cryo-electron microscopy (cryo-EM), we determine the structures of the LGR4 and LGR4-RSPO2 and LGR4-RSPO2-ZNRF3 complexes, in which LGR4 and ZNRF3 include their ECD and TMD. These structures reveal the overall architecture of LGR4 and the 2:2:2 organization of the ternary complex, which may provide a basis for understanding how LGR4 and RSPO relieve ZNRF3-mediated suppression of Wnt/β-catenin signaling.

## Results

### The overall structure of LGR4

We purified lauryl maltose neopentyl glycol/ cholesteryl hemi-succinate (LMNG/CHS) detergent-solubilized human LGR4 (residues 1-822) containing ECD and TMD and determined its structure using cryo-EM to a resolution of 3.5 Å (Fig. 1a, b) (Supplementary Table 1 and Supplementary Fig. 5). In the cryo-EM map, the ECD (residues 29-525), TMD (residues 539-822), and connecting segment between the ECD and TMD (residues 526-538) were observed and modeled, except for the extended-loop region at the end of the ECD (residues 476-518).

LGR4 ECD adopts a typical horseshoe structure comprising 17 LRRs (residues 58-455) flanked by N-terminal (LRRNT, residues 29-57) and C-terminal (LRRCT, residues 456-525) caps (Fig. 1b). The β-sheet structure on the concave surface of the LGR4 LRR was disrupted at the LRR10-LRR11 junction. Thus, the LGR4 LRR can be divided into two structural segments, LRR1-10 and LRR11-17, with slightly different orientations (Fig.1b). LGR4 TMD comprises 7-transmembrane (TM) helices typical of GPCRs. LGR4 TMD can be superimposed well on the inactive conformation of the luteinizing hormone-choriogonadotropin receptor (LHCGR/LGR2) with a root-mean-square deviation (RMSD) value of 1.08 Å (Fig. 1c)[16]. Therefore, the GPCR 7-TM structure has an inactive conformation.

The ECD was tilted to about 45° relative to the membrane layer, a configuration maintained by interactions at the hinge region between the ECD and TMD, comprising the LRRCT and the following connecting segment (Fig. 1b, d). The hinge-mediated ECD-TMD interface comprised two layers of interactions (Fig. 1d). In the upper layer, the LRRCT helix packs complementarily against the extracellular loop (ECL)1 helix from the TMD and the adjacent connecting segment, with the Y468 side chain inserted into a small pocket formed between the ECL1 helix and the connecting segment (Fig. 1d). In addition, a C471-C532 disulfide bond was formed between the LRRCT helix and the connecting segment. In the lower layer, the connecting segment formed a three-successive-turn structure that interacted extensively with the presumed ligand-binding pocket on the extracellular side of the TMD. The first turn (residues 527-530) connects with ECL1, ECL2, and TM7, and the F529 side chain interacts with F696 (ECL2) and M782 (TM7) (Fig. 1d). The second turn (residues 530-534) interacts with the loop region preceding the ECL1 and LRRCT helices. The third turn (residues 534-537) folds back toward the TMD pocket, where two Leucine residues (L535 and L536) occupy the space between TM1, TM2, ECL1, ECL2, and TM7, forming extensive hydrophobic interactions with the residues from these regions (Fig. 1d).

### LGR4 undergoes no conformational changes upon binding to RSPO2

We purified the binary complex of human LGR4 and RSPO2 using gel-filtration chromatography from a mixture of individually purified LGR4 and RSPO2 (residues 20-143, Fu1-2 domains, Fig. 1a). We determined its

structure using cryo-EM at an overall resolution of 4.0 Å (Fig. 2; Supplementary Figs. 6 and 7 and Supplementary Table 1). RSPO2 Fu1-2 domains exhibited an elongated form comprising multiple β-hairpins stabilized by disulfide bonds (Supplementary Fig. 7a). Consistent with the previously determined crystal structure of LGR4 ECD in complex with RSPO1, RSPO2 binds and crosses the N-terminal concave surface of LGR4 ECD LRR3-9 with 1:1 stoichiometry (Fig. 2a)[21] (Supplementary Fig. 7b). The LGR4-RSPO2 interface, primarily mediated by hydrophobic and electrostatic interactions, is mostly conserved between the LGR4-RSPO2 and LGR4-RSPO1 complexes, as expected due to the high sequence similarity among RSPO1-4 (Supplementary Fig. 3).

RSPO2-bound LGR4 structure is almost identical to the apo LGR4 structure with an overall RMSD of 0.40 Å, indicating that RSPO2 binding does not induce any conformational differences in the LGR4 structure (Fig. 2b). This is in sharp contrast to the 'push-pull' activation mechanism of related LGR1-3, in which hormone bound to the receptor sterically clashes with the membrane, causing an upward movement of the ECD to avoid the clash, resulting in receptor activation (Supplementary Fig. 7c).

### LGR4 and RSPO2 enclose ZNRF3 dimer in the LGR4-RSPO2-ZNRF3 ternary complex

We purified LMNG/CHS detergent-solubilized ternary complexes of human LGR4, RSPO2, and ZNRF3 (residues 56-267, ECD-TMD) from Expi293F cells cotransfected with LGR4, RSPO2, and ZNRF3 (Supplementary Fig. 8a). Cryo-EM analysis of this sample revealed two distinct LGR4−RSPO2−ZNRF3 complexes with stoichiometric ratios of 1:1:2 and 2:2:2, corresponding to a heterotetramer and a heterohexamer, respectively (Supplementary Fig. 8b). We obtained cryo-EM maps of LGR4-RSPO2-ZNRF3$_{1:1:2}$ heterotetramer and LGR4-RSPO2-ZNRF3$_{2:2:2}$ heterohexamer to resolutions of 3.2 Å and 3.5 Å, respectively (Fig. 3a, b; Supplementary Figs. 8b, c and Supplementary Table 1). In both complexes, the cryo-EM densities of the TMD regions were relatively weak compared to those of the extracellular regions (Supplementary Fig. 8b).

In the LGR4-RSPO2-ZNRF3$_{1:1:2}$ and LGR4-RSPO2-ZNRF3$_{2:2:2}$ complexes, the ECD of ZNRF3 existed as a dimer with the same configuration as the previously reported crystal structures of ZNRF3 ECD and its complex with RSPO (Fig. 3a, b and Supplementary Fig. 9a). In the LGR4-RSPO2-ZNRF3$_{1:1:2}$ complex, RSPO2 was bound to one protomer of the ZNRF3 dimer through which LGR4 was associated, forming a 1:1:1 complex on one side of the structure (Fig. 3a). In the LGR4-RSPO2-ZNRF3$_{2:2:2}$ complex, another copy of the LGR4*-RSPO2* complex was bound to the other protomers of the ZNRF3 dimer, ZNRF3* (the asterisk indicates the second protomer in the dimer), forming another set of 1:1:1 complexes, thus completing the C2-symmetrical 2:2:2 complex (Fig. 3b). The 2:2:2 complex was generally consistent with the previously predicted 2:2:2 model[26], created by superposing two LGR4-RSPO2 complexes onto each RSPO2 protomer of the ZNRF3-RSPO2 complex, as well as the low-resolution crystal structure of the LGR5$_{ECD}$-RSPO2-ZNRF3$_{ECD}$ 2:2:2 complex[25] (Supplementary Fig. 9b).

RSPO2 plays a major role in the 1:1:1 assembly by interacting with LGR4 and ZNRF3: RSPO2 binds LGR4 through the Fu1-2 domain and, on the opposite side, connects to ZNRF3 primarily through the Fu1 domain (Fig. 3b). The LGR4-RSPO2 and RSPO2-ZNRF3 interfaces were essentially identical to those in the LGR4-RSPO2 complex in this study and those observed previously, respectively[8,14,21,22] (Supplementary Figs. 9c, d).

No apparent interactions occurred between the ECDs of ZNRF3 and LGR4 in the 1:1:1 assembly. However, ZNRF3 made contact with LGR4*, thereby stabilizing the overall complex, particularly ZNRF3 dimerization (Fig. 3b, c). This interaction has not been predicted or observed previously[24,25]. The newly identified ZNRF3-LGR4* interface comprises the edge of the ZNRF3 ECD β-hairpin that mediates ZNRF3 dimerization and the lateral surface of LRR15-17 and LRRCT of LGR4

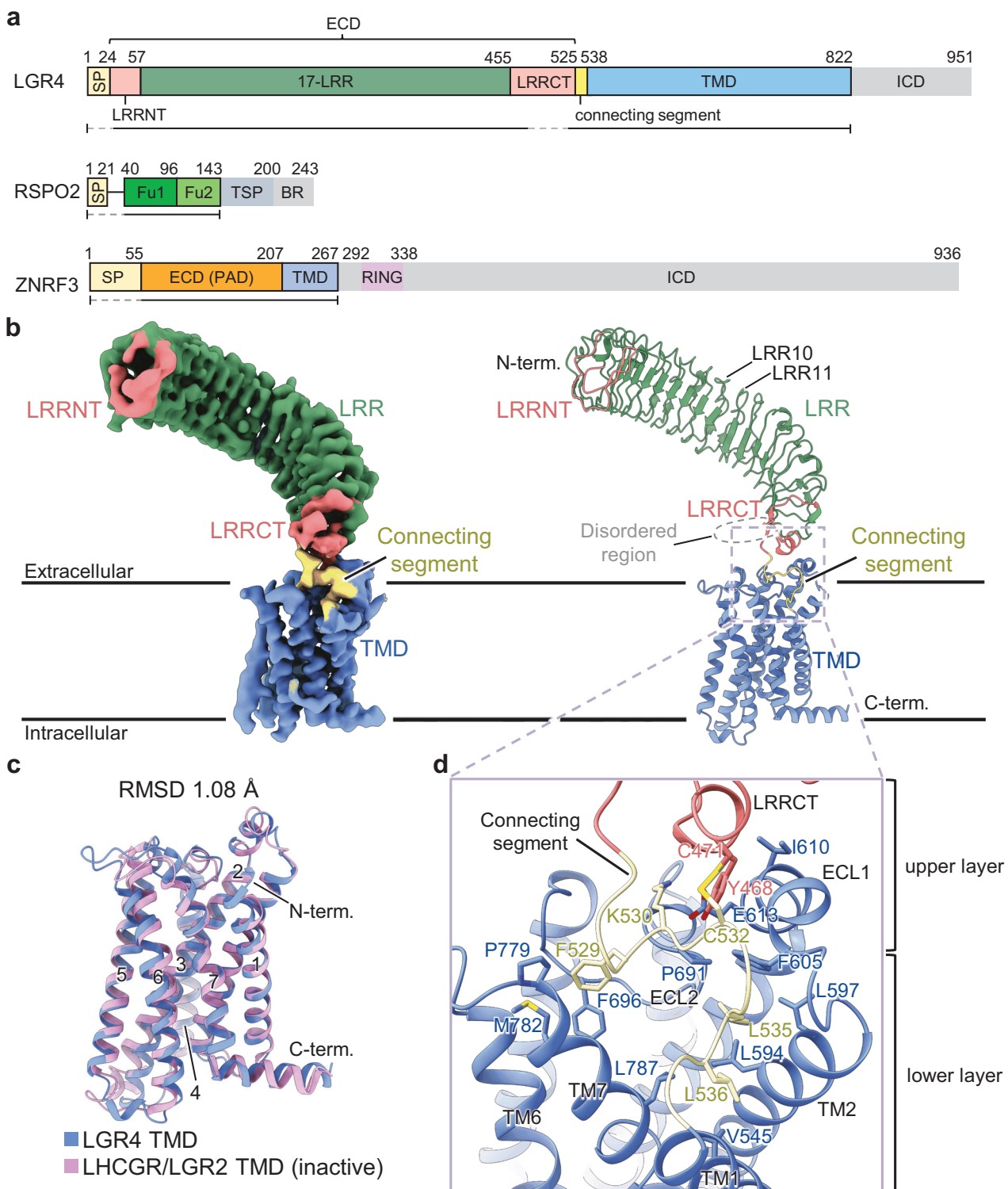

**Fig. 1 | Structure of LGR4. a** Schematic diagrams of the domain organization of human LGR4, RSPO2, and ZNRF3. SP signal peptide, LRR Leucine-rich repeat, LRRNT LRR N-terminal motif, LRRCT LRR C-terminal motif, ECD extracellular domain, TMD transmembrane domain, ICD intracellular domain, Fu furin domain, TSP thrombospondin-like, BR basic amino acid-rich, PAD protease-associated domain, RING RING domain. The regions used in this study are shown below the diagram and the dashed lines represent the unmodeled regions. **b** Overall structure of human LGR4. The cryo-EM map (left) and ribbon model (right) are shown. LRRNT

(salmon pink), 17-LRR (green), LRRCT (salmon pink), connecting segments (yellow), and TMD (blue) are shown in different colors. The gray dashed lines indicate the disordered loop region within the LRRCT. **c** Structural comparison of the 7-TM structures of LGR4 TMD (blue) and the inactive conformation of LHCGR/LGR2 TMD (pink, PDB: 7FIJ) with the labeled TM numbers. **d** Close-up view of the ECD-TMD hinge region (dashed rectangle in (**b**)). ECL extracellular Loop, TM Transmembrane helix.

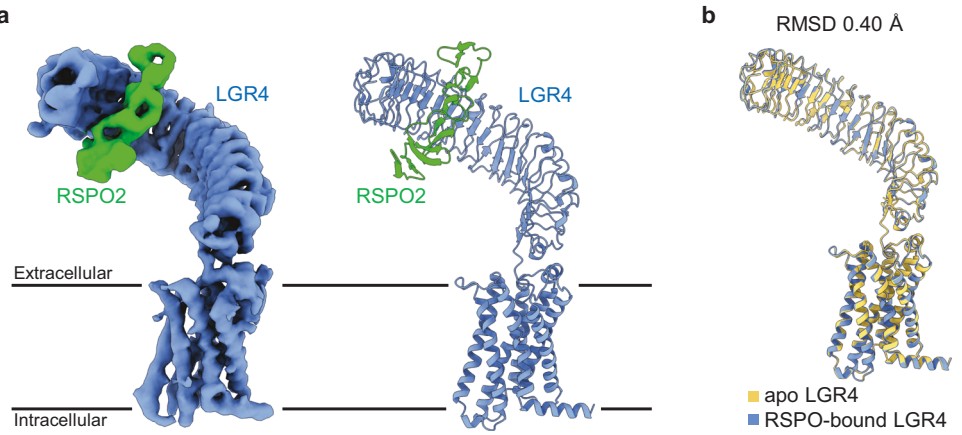

**Fig. 2 | Structure of the LGR4-RSPO2 complex. a** Overall structure of the LGR4-RSPO2 complex. Cryo-EM map (left) and ribbon models (right) with LGR4 in blue and RSPO2 in green. **b** Superposition of apo LGR4 (yellow) and RSPO2-bound LGR4 (blue). RMSD: Root Mean Square Deviation.

(Fig. 3b, d). At the center of the interface, ZNRF3 Y77 and T78 formed hydrogen bonds with G435 and N437 of LGR4*, respectively (Fig. 3c). In addition, two pairs of electrostatically complementary residues (E67 of ZNRF3 and R392 of LGR4* and E62 of ZNRF3 and R460 of LGR4*) located at the peripheral regions of the interface might support this interaction. To investigate the functional significance of this interface, we designed a truncated LGR4 variant lacking the ZNRF3-binding region, LRR15-LRRCT domain (Δaa 390-465), and assessed its activity in Wnt/β-catenin signaling using the Wnt/β-catenin reporter TOPFlash system in human embryonic kidney (HEK)293 T cells[27]. The LGR4 (ΔLRR15-LRRCT) mutant exhibited significantly reduced Wnt/β-catenin signaling activity (Fig. 3d and Supplementary Fig. 10a, b), despite being expected to retain RSPO2 binding, suggesting that the newly identified extracellular interface between ZNRF3 and LGR4 is important for Wnt/β-catenin signaling potentiation. Notably, this interface appears to facilitate the preassembly of LGR4 and ZNRF3 into a low-affinity complex at the cell membrane before RSPO engagement (Fig. 3e). RSPO binding then serves as a molecular bridge, stabilizing the interactions and thereby inducing the formation of a more stable ternary complex (Fig. 3e). The intermolecular LGR4-RSPO2, RSPO2-ZNRF3, and ZNRF3*-LGR4 (or ZNRF3-LGR4*) interfaces at the extracellular region were almost identical between the LGR4-RSPO2-ZNRF3$_{1:1:2}$ and LGR4-RSPO2-ZNRF3$_{2:2:2}$ complexes (Supplementary Fig. 9e).

In the LGR4-RSPO2-ZNRF3$_{1:1:2}$ structure, the TM helix of only one protomer, ZNRF3*, was defined by the cryo-EM density, whereas that of the other protomer, ZNRF3, was not visible, indicating that the TM helices of ZNRF3 are flexible (Fig. 3a and Supplementary Fig. 8b, c). The observed TM helix of ZNRF3* was aligned and in contact with TM1 and TM7 of LGR4 on the extracellular side of the membrane but made no contact on the intracellular side (Fig. 3f).

In the LGR4-RSPO2-ZNRF3$_{2:2:2}$ complex, the TM helices of ZNRF3 protomers were visible in the cryo-EM density (Fig. 3b and Supplementary Fig. 8b, d). The map quality was insufficient to model the TM helices precisely; however, it was sufficient to locate the TM positions in the complex. The two TM helices of ZNRF3 were loosely sandwiched between the two TMDs of LGR4, especially between TM1 and TM7 of one LGR4 promoter and TM6 of the other LGR4* promoter (Fig. 3g). The two TM helices of ZNRF3 made contact with each other at a distance of 5–7 Å on the extracellular side of the membrane (Fig. 3b). In addition to the ECD dimerization of ZNRF3, these intermolecular contacts in the membrane restrict the spatial positioning of the TM helices of ZNRF3 and bring the intracellular region of ZNRF3 into proximity, which may facilitate the auto-ubiquitination of ZNRF3[20,28,29] (Fig. 3b, g). When the TMD of LGR4 (7-TM, aa526-822) was replaced with that of CD4 (1-TM) according to the previous study[30], the resulting LGR4 variant (CD4 TMD) showed nearly a 50% reduction in RSPO2-induced Wnt/β-catenin signaling activity compared to the wild-type LGR4 (Fig. 3h and Supplementary Figs. 10a, b). These results suggest that the spatial constraints of the TM helices of ZNRF3 imposed by the 7-TM TMD of LGR4 may be important for the potentiation of Wnt/β-catenin signaling.

The ECD and TMD structures of LGR4 did not differ among the apo LGR4, LGR4-RSPO2-ZNRF3$_{1:1:2}$, and LGR4-RSPO2-ZNRF3$_{2:2:2}$ complexes; the relative orientations between the ECD and TMD were slightly variable, possibly due to the LGR4-ZNRF3 interfaces formed in the ECD and TMD regions of the ternary complexes (Supplementary Fig. 9f).

## The functional importance of the ZNRF3 dimerization through complexation with LGR4 and RSPO2

In the context of LGR4, RSPO2, and ZNRF3 axis, previous studies have suggested that ligand-induced dimerization of ZNRF3 might be important for its membrane clearance via endocytosis[29,31]. Therefore, we evaluated the functional significance of ZNRF3 dimerization in the 2:2:2 assembly. We utilized a dimerization-deficient mutant of ZNRF3 (E95N/E97T) (Supplementary Fig. 10c, d), previously confirmed to be incapable of dimerization[22], and tested the dimerization ability of ZNRF3 with or without LGR4 and RSPO2 by pull-down assay (Fig. 4a). The results showed that although ZNRF3 dimerization occurred in the absence of LGR4 and RSPO2, LGR4/RSPO2 co-expression markedly enhanced ZNRF3 dimerization. In contrast, the ZNRF3 (E95N/E97T) mutant exhibited severely impaired dimerization, even in the presence of LGR4 and RSPO2 (Fig. 4a). This finding is consistent with previous in vitro studies[22], both demonstrating that ZNRF3 dimerization is strongly enhanced upon RSPO and LGR4 engagement.

We next assessed the activity of wild-type and mutant ZNRF3 in Wnt/β-catenin signaling. ZNRF3 (E95N/E97T) exhibited a significant decrease in the TOPFlash reporter activity (Fig. 4b and Supplementary Figs. 10c, d). These findings further support that ZNRF3 dimerizes through the formation of a 2:2:2 complex with LGR4 and RSPO2, which is critical for the potentiation of Wnt/β-catenin signaling.

## Discussion

A structural analysis of LGR4 and its complex with RSPO2 and ZNRF3 suggests a distinct G-protein-independent signaling mechanism for LGR4 (Fig. 5). In the ternary complex, LGR4 serves as an engagement receptor to recruit ZNRF3 by interacting with RSPO2, whereas ZNRF3 functions as an effector receptor to initiate subsequent signaling[27]. While the interactions between RSPO2 and ZNRF3, as well as between LGR4 and RSPO2, have been well characterized, our study further

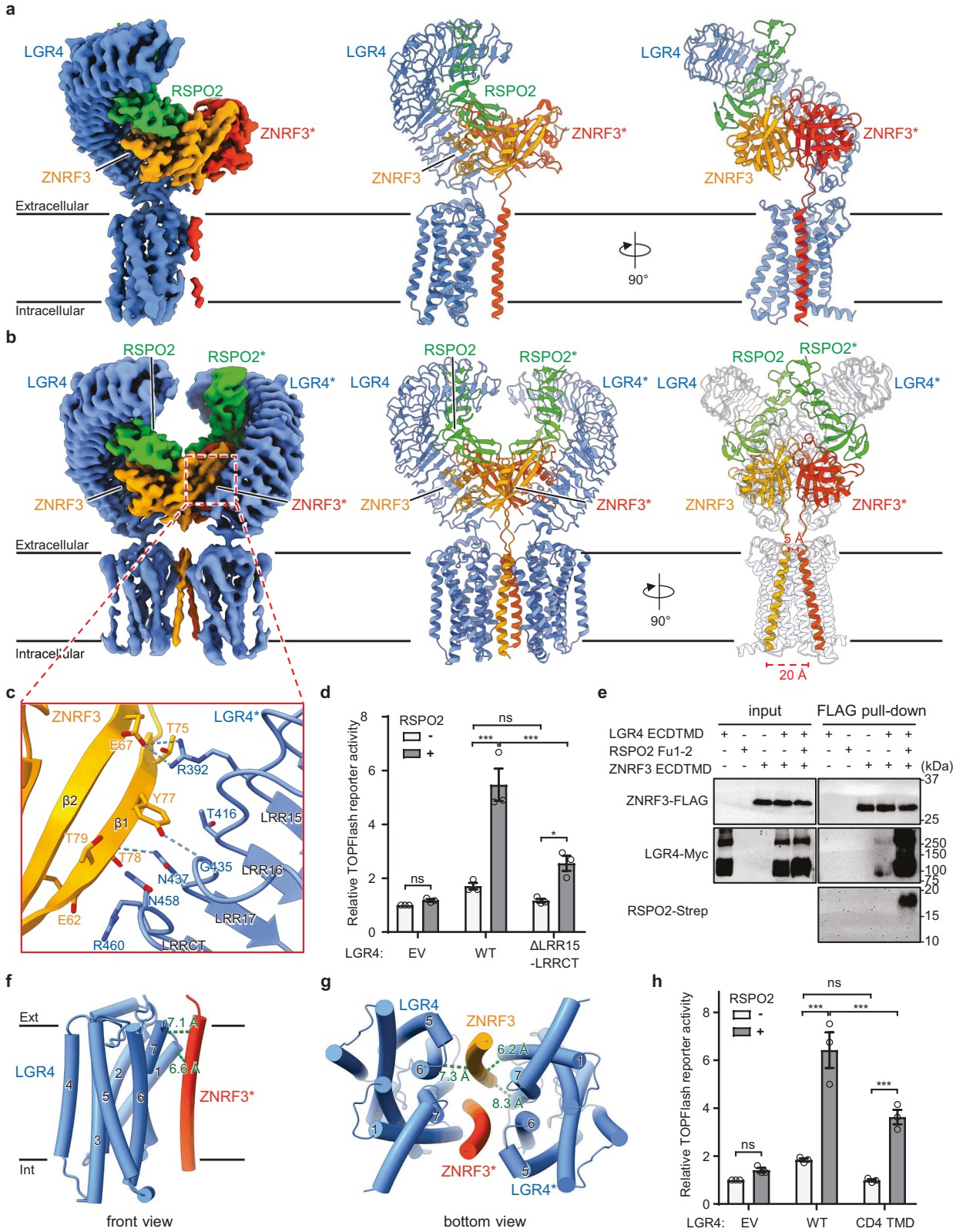

revealed that LGR4 also establishes direct contact with ZNRF3 (Fig. 3b–e). Each component in the complex engages with the other two, thereby cooperatively stabilizing the overall arrangement of the 2:2:2 heterohexamer and promoting ZNRF3 dimerization required for the potentiation of Wnt/β-catenin signaling. In the 2:2:2 heterohexamer configuration, ZNRF3 ECD and TMD are largely enclosed by LGR4 and RSPO2, resulting in steric hindrance that prevents ZNRF3

from binding to the FZDs, thereby suppressing its ubiquitination activity. Moreover, the enhanced ZNRF3 dimerization in this configuration, along with the restricted movement of the ZNRF3 TMD, may promote sustained self-ubiquitination and subsequent membrane clearance through endocytosis. This notion is consistent with a previous report[28] and is further supported by our functional data (Fig. 4). In either case, ZNRF3 undergoes inactivation and endocytosis, leading

**Fig. 3 | Structure of the LGR4-RSPO2-ZNRF3 complex. a** Overall structure of the LGR4-RSPO2-ZNRF3$_{1:1:2}$ complex. The cryo-EM map (left) and ribbon models (middle and right) with LGR4 (blue), RSPO2 (green), and ZNRF3 (orange/red). The asterisk indicates a second protomer in the dimer. **b** Overall structure of the LGR4-RSPO2-ZNRF3$_{2:2:2}$ complex. The cryo-EM map (left) and the ribbon models (middle and right). **c** Close-up view of the interface between ZNRF3 and LGR4 (ZNRF3-LGR4*) in the extracellular region. The dashed lines indicate salt bridges and hydrogen bonds. **d** TOPflash reporter assays using full-length wild-type (WT) or mutant (ΔLRR15-LRRCT) LGR4. HEK293T cells were transfected with siRNA against LGR4 and LGR5, followed by plasmid transfection, and then treated with 5% Wnt3a-conditioned medium in the presence or absence of 3 ng/mL RSPO2. EV: empty vector. $n = 3$ biological replicates. Bars represent mean ± standard error of the mean (SEM), and dots show individual data points. Statistical significance was determined using two-way ANOVA with two-sided Tukey's test. *$P = 0.038$, ***$P < 0.001$. ns not significant. **e** Pull-down assay of LGR4 and ZNRF3 in the

presence or absence of RSPO2. Expi293F cells were transfected with ZNRF3 ECDTMD (Flag-tagged), LGR4 ECDTMD (Myc-tagged), and RSPO2 Fu1-2 (Strep-tagged). Cell lysates were incubated with anti-FLAG affinity beads, and bound proteins were detected by western blotting. Data are representative of three independent experiments. **f** The TMD region of the LGR4-RSPO2-ZNRF3$_{1:1:2}$ complex (front view). The TM helix of ZNRF3* is in contact with TM1 and TM7 of LGR4, and the distances between them are indicated. **g** The TMD region of the LGR4-RSPO2-ZNRF3$_{2:2:2}$ complex (bottom view). The two TM helices of ZNRF3 are sandwiched between TM1, TM6, and TM7 of LGR4, and the distances between them are indicated. **h** TOPflash reporter assays using full-length wild-type (WT) or mutant (CD4 TMD) LGR4. HEK293T cells were stimulated as in (**d**). $n = 3$ biological replicates. Bars represent mean ± SEM, and dots show individual data points. Statistical significance was determined using two-way ANOVA with two-sided Tukey's test. ***$P < 0.001$. ns, not significant. Source data are provided as a Source data file.

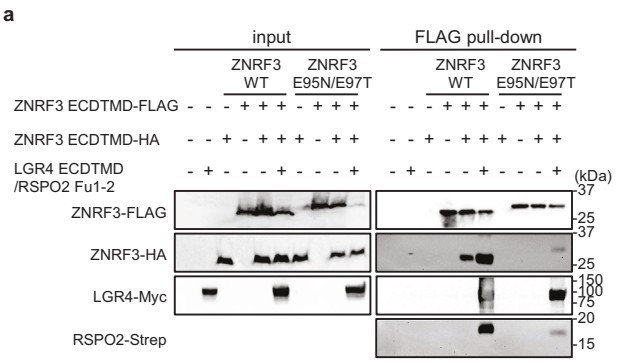
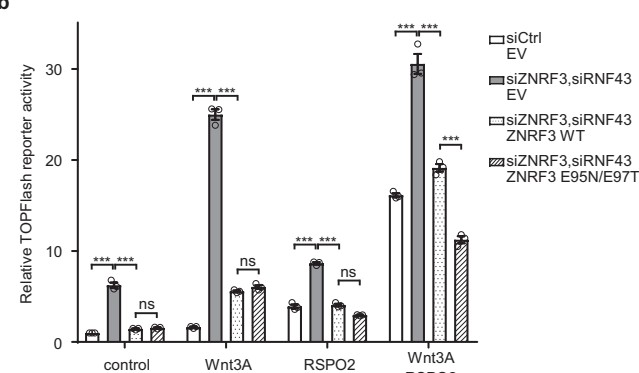

**Fig. 4 | ZNRF3 dimerization through complexation with LGR4 and RSPO2 is important for Wnt/β-catenin signal potentiation. a** Pull-down assay of ZNRF3 (WT) or ZNRF3 (E95N/E97T) in the presence or absence of LGR4 and RSPO2. Expi293F cells were transfected with ZNRF3 ECD-TMD (Flag-tagged), ZNRF3 ECD-TMD (HA-tagged), LGR4 ECD-TMD (Myc-tagged), and RSPO2 Fu1-2 (Strep-tagged). Cell lysates were incubated with anti-FLAG affinity beads, and bound proteins were detected by western blotting. Data are representative of three independent experiments. **b** TOPflash reporter assays using full-length wild-type (WT) or

dimerization-deficient mutant (E95N/E97T) ZNRF3. HEK293T cells were transfected with the indicated siRNA, followed by plasmid transfection, and then treated with or without 5% Wnt3a-conditioned medium in the presence or absence of 200 ng/mL RSPO2. EV: empty vector. $n = 3$ biological replicates. Bars represent mean ± standard error of the mean (SEM), and dots show individual data points. Statistical significance was determined using Two-way ANOVA with two-sided Tukey's test. ***$P < 0.001$. ns not significant. Source data are provided as a Source data file.

to the deregulation of its inhibitory effect on FZDs and LRP5/6, thereby enhancing Wnt/β-catenin signaling. In addition to affecting ZNRF3 ubiquitination activity and promoting its membrane clearance, increasing evidence suggests that LGR4/5 and RSPO may also regulate Wnt/β-catenin signaling by directly interacting with Wnt receptors, independent of ZNRF3[32–34]. Moreover, RSPO2 and RSPO3 were found to interact with ZNRF3 to potentiate Wnt/β-catenin signaling even in the absence of LGRs[35]. However, further research is required to clarify these interactions and their specific contributions to Wnt/β-catenin signaling.

The mechanism by which LGR4- and RSPO-induced ZNRF3 dimerization potentiates Wnt/β-catenin signaling has not been fully elucidated. Receptor tyrosine kinases undergo ligand-induced dimerization, triggering trans-autophosphorylation and subsequent activation[36–38]. Similarly, ZNRF3 dimerization may facilitate its self-ubiquitination and regulatory function, thereby enhancing the Wnt/β-catenin signaling. Future structural investigations of ligand-induced conformational changes in full-length ZNRF3 may clarify the underlying mechanism.

Although full-length LGR4 has been reported to exist as dimers on the cell surface[26,28], no dimerized form of LGR4 alone was observed by gel chromatography or cryo-EM analysis, possibly due to its relatively weak dimerization (Supplementary Figs. 5a, b). Moreover, no apparent contact was observed between the two LGR4 protomers within the 2:2:2 heterohexamer. Therefore, it remains unclear whether the

preformed LGR4 dimer adopts a configuration distinct from that observed in the heterohexamer.

In LGR2 and LGR3, in addition to their cognate hormone ligands, synthetic allosteric agonists have been identified that target a hydrophobic pocket located on the top half of the TMD extracellular side, corresponding to the orthosteric ligand-binding site in many other GPCRs[16,17]. These agonists, either alone or in combination with hormonal ligands, induce the active conformation of TMD. To date, no such ligand has been identified for LGR4[39–42]; however, LGR4 features a similar hydrophobic pocket located on its TMD, comprising residues from TM3, TM5, TM6, and TM7, as well as residues from ECL2 and ECL3 (Supplementary Fig. 11). Notably, most of the residues forming this pocket were conserved among LGR1-6 (Supplementary Fig. 2). In the inactive conformation of LGR4, F529, F696, and M782 blocked the entrance to the pocket (Supplementary Fig. 11b); however, the structural transition from the inactive to the active state may allow the pocket to widen, as observed in LGR2, enabling it to accommodate such ligands (Supplementary Fig. 11c).

In conclusion, this study elucidated detailed interactions between LGR4, ZNRF3, and RSPO2, along with their complex organizational mechanisms, providing insights into LGR4-mediated potentiation of Wnt/β-catenin signaling. During the review process of this manuscript, the structures of the LGR4-RSPO2-ZNRF3 complexes were reported. These findings are highly complementary to our study and further strengthen our conclusions[43].

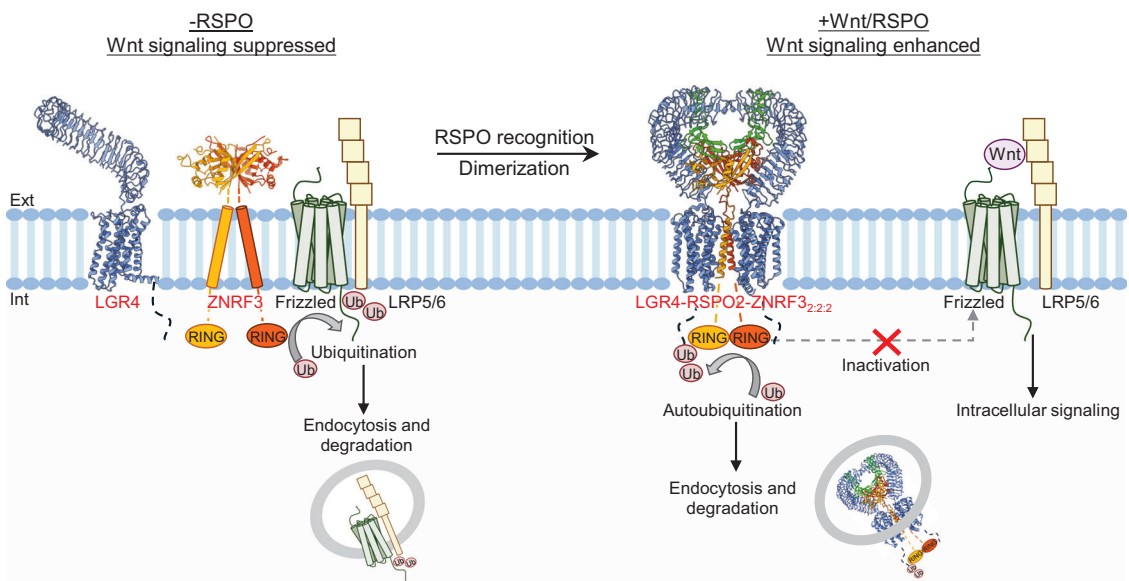

**Fig. 5 | Mechanism of Wnt/β-catenin signaling regulation by LGR4, R-spondin, and ZNRF3.** In the absence of RSPO (-RSPO), ZNRF3 ubiquitinated Frizzled and LRP5/6, promoting their endocytosis and degradation, thereby suppressing Wnt/β-catenin signaling. LGR4, together with RSPO (+RSPO), recruits ZNRF3, leading to LGR4-RSPO-ZNRF3$_{2:2:2}$ formation. Formation of the ternary complex brings two ZNRF3 molecules into proximity, facilitating the auto-ubiquitination and subsequent endocytosis of ZNRF3. In addition, the formation of the ternary complex may sterically restrict ZNRF3 from interacting with Frizzled and LRP5/6 for ubiquitination. Consequently, the ZNRF3-mediated suppression of Frizzled and LRP5/6 receptors was relieved, leading to enhanced Wnt/β-catenin signaling. Structure model of ZNRF3 ECD based on crystal structures of ZNRF3 (orange and orange red, PDB: 4C8C). Ext extracellular, Int intracellular.

## Methods

### Expression and purification of LGR4

The gene encoding human LGR4 (residue 1-822, UniProt accession number Q9BXB1, Fig. 1a) was purchased (RIKEN DNA Bank, #HKR247353) and cloned with C-terminal tobacco etch virus (TEV) protease cleavage sequence and FLAG-His10 tags into the pFastBac Dual vector (Thermo Fisher Scientific). Recombinant baculovirus was produced using ExpiSf9 cells (Thermo Fisher Scientific, A35243) cultured in ExpiSf CD medium (Thermo Fisher Scientific), following the manufacturer's instructions (Thermo Fisher Scientific). For protein expression, ExpiSf9 cells were infected with recombinant baculoviruses and incubated for 60–70 h at 27 °C. The cells were collected via centrifugation and lysed by sonication in a buffer containing 25 mM Tris·HCl (pH 7.5), 0.20 M NaCl, 5% (w/v) glycerol, and 1% protease inhibitor cocktail (Nacalai Tesque). After removal of cell debris via centrifugation at 5000 × $g$ for 10 min, the membrane fraction was collected via ultracentrifugation at 180,000 × $g$ for 1 h and was solubilized in a buffer containing 25 mM Tris·HCl (pH 7.5), 0.20 M NaCl, 1% LMNG (Anatrace), and 0.1% CHS (Sigma-Aldrich) for 1 h at 4 °C. After removing insoluble materials via centrifugation at 48,000 × $g$ for 10 min, the solubilized proteins were incubated with Ni-NTA agarose (FUJIFILM Wako) for 1 hour at 4 °C. The resin was washed with >20 column volume of wash buffer (25 mM Tris·HCl, pH 7.5, 0.20 M NaCl, 25 mM imidazole, 0.01% GDN). Proteins were eluted using a buffer containing 25 mM Tris·HCl, pH 7.5, 0.20 M NaCl, 300 mM imidazole, and 0.01% GDN and further purified via size-exclusion chromatography (SEC) (Superose 6 10/300 GL, Cytiva) in a buffer containing 25 mM HEPES-NaOH, pH 7.5, 0.15 M NaCl, and 0.01% GDN. Fractions containing LGR4 were collected and concentrated using an Amicon Ultra centrifugal filter (100-kDa molecular weight cut-off).

### Expression and purification of LGR4-RSPO2 complex

The codon-optimized DNA fragment encoding human RSPO2 (residue 1-143, UniProt accession number Q6UXX9, Fig. 1a) was synthesized (Thermo Fisher Scientific) and cloned with C-terminal tobacco etch virus (TEV) protease cleavage sequence and His10 tags into the pEZT-BM vector[44]. Expi293F cells (Thermo Fisher Scientific, A14635) cultured in Expi293 Expression Medium (Thermo Fisher Scientific) were transfected with the vector DNA/polyethylenimine complex at a cell density of approximately 3 × 10⁶ cells/mL and incubated at 37 °C under 8% CO$_2$ with agitation at 120 r.p.m. At 20–24 h after transfection, 10 mM sodium butyrate was added to the expression medium and further incubated at 30 °C for 3 days. The proteins secreted into the culture medium were purified by Ni-NTA resin (FUJIFILM Wako) and further purified via SEC (Superose 6 10/300 GL, Cytiva) in a buffer containing 25 mM HEPES-NaOH (pH 7.5), and 0.15 M NaCl. Fractions containing RSPO2 were collected and concentrated using an Amicon Ultra centrifugal filter (10-kDa molecular weight cut-off). To obtain the LGR4-RSPO2 complex, purified LGR4 was incubated with RSPO2 at a 1:2 molar mass ratio for 3 hours at 4 °C, followed by purification using SEC (Superose 6 Increase 10/300 GL, Cytiva) in a buffer containing 25 mM HEPES-NaOH, pH 7.5, 0.15 M NaCl, and 0.01% GDN. Fractions containing the LGR4-RSPO2 complex were collected and concentrated using an Amicon Ultra centrifugal filter (100-kDa molecular weight cut-off).

### Expression and purification of LGR4-RSPO2-ZNRF3 complex

The gene encoding human LGR4 (residues 1–822, UniProt accession number Q9BXB1) was cloned with C-terminal TEV protease cleavage sequence and FLAG-His10 tags into the pEZT-BM vector. The condon-optimized DNA fragment encoding human ZNRF3 (residues 56–267, UniProt accession number Q9ULT6, Fig. 1a) was synthesized (Thermo Fisher Scientific) and cloned into the pEZT-BM vector with the rabbit immunoglobulin (Ig) heavy chain V signal peptide at the N-terminus. Expi293F cells were co-transfected with the LGR4, RSPO2, and ZNRF3 vectors at a cell density of approximately 3 × 10⁶ cells/mL and incubated at 37 °C under 8% CO$_2$ with agitation at 120 r.p.m. At 20–24 h after transfection, 10 mM sodium butyrate was added to the expression medium and further incubated at 30 °C for 2 days. The cells were collected and sonicated in a buffer containing 25 mM Tris·HCl (pH 7.5), 0.20 M NaCl, 5% (w/v) glycerol, and 1% protease inhibitor cocktail (Nacalai Tesque). After removal of cell debris via centrifugation at 5000 × $g$ for 10 min, the membrane fraction was collected via ultracentrifugation at 180,000 × $g$ for 1 h and was solubilized in a buffer containing 25 mM Tris·HCl (pH 7.5), 0.20 M NaCl, 1% LMNG (Anatrace),

and 0.1% CHS (Sigma-Aldrich) for 1 h at 4 °C. After removing insoluble materials via centrifugation at $48{,}000 \times g$ for 10 min the solubilized proteins were purified by Anti-DYKDDDDK affinity resin (FUJIFILM Wako) and further purified via SEC (Superose 6 10/300 GL, Cytiva) in a buffer containing 25 mM HEPES-NaOH (pH 7.5), 0.15 M NaCl, and 0.01% GDN. Fractions containing the LGR4-RSPO2-ZNRF3 complex were collected and concentrated using an Amicon Ultra centrifugal filter (100-kDa molecular weight cut-off).

## Cryo-EM grid preparation and data collection

Protein samples were diluted to final concentrations of 4.0–8.0 mg/ml in SEC buffer (25 mM HEPES-NaOH, pH 7.5, 0.15 M NaCl, 0.01 % GDN). A 3-μL aliquot was applied to freshly glow-discharged Quantifoil holey carbon grids (R1.2/1.3, Cu, 300 mesh). After 2.0–3.0 s of blotting in 100% humidity at 6 °C with blot force 10, the grid was plunged into liquid ethane using a Vitrobot Mark IV (Thermo Fisher Scientific). Cryo-EM micrographs were obtained using a Titan Krios G4 microscope (Thermo Fisher Scientific) running at 300 kV and equipped with a Gatan Quantum-LS Energy Filter (GIF) and a Gatan K3 camera in the electron counting mode at the Cryo-EM facility in the University of Tokyo. Imaging was performed at a nominal magnification of ×105,000, corresponding to a calibrated pixel size of 0.83 Å per pixel. Typically, each movie was recorded for 1.6 or 2.0 s and subdivided into 48 or 60 frames with an accumulated exposure of about 48 or 60 e⁻/Å² at the specimen. Movies were acquired by fast acquisition mode using the EPU software (Thermo Fisher Scientific) with defocus ranges of −1.0 to −2.0 μm.

## Cryo-EM image processing and model building

The cryo-EM datasets were processed using cryoSPARC (v4.4.0)[45]. Raw movie stacks were corrected with patch motion correction, and the contrast transfer function (CTF) parameters were estimated using Patch CTF estimation. Particles were picked using a blob picker or template picker, and 2D classification and heterogeneous refinement were performed to select good-quality particles for 3D map reconstruction. Final 3D maps were reconstructed by non-uniform refinement with Global CTF refinement. The final resolution was estimated using the gold-standard Fourier shell correlation (FSC) between two independently refined half maps (FSC = 0.143).

For model building, the crystal structure of LGR4 ECD (PDB: 4KT1) and the AlphaFold2 models of LGR4, RSPO2, and ZNRF3 were docked into the cryo-EM density map using Chimera[46], followed by iterative adjustment and rebuilding in Coot[47]. Real-space refinements were performed using PHENIX programs[48]. The model was validated using the comprehensive validation (cryo-EM) module in PHENIX[48]. The cryo-EM maps and the atomic coordinates have been deposited into the Electron Microscopy Data Bank (EMDB) and the Protein Data Bank (PDB), respectively. Data collection and structural refinement statistics are listed in Supplementary Table 1. Structure figures were prepared in Chimera and Chimera X[49].

## Antibodies

Anti-DDDDK-tag mAb-HRP-DirecT (anti-FLAG, 1:2500 dilution; Medical and Biological Laboratories, M185-7), anti-Myc (1:2500 dilution; Medical and Biological Laboratories, 192-3), anti-Strep II (1:2000 dilution; Medical and Biological Laboratories, M211-3), anti-HA tag (1:2500 dilution; Medical and Biological Laboratories, M180-3), anti-β-actin (1:2500 dilution; Santa Cruz Biotechnology, sc-47778, Lot no. J0421), and rabbit anti-mouse IgG H&L (HRP) (1:2500 dilution; Abcam, ab6728) were used for western blotting.

## Pull-down assay

Flag-tagged ZNRF3 ECD-TMD, HA-tagged ZNRF3 ECD-TMD, Myc-tagged LGR4 ECD-TMD, Strep-tagged RSPO2 Fu1-2 were co-expressed in Expi293F cells. Cells were harvested and lysed in a buffer containing 25 mM Tris-HCl (pH 7.5), 0.15 M NaCl, 10% glycerol, 1% LMNG/0.1% CHS. After centrifugation, the soluble supernatant was incubated with 20 μL anti-DYKDDDDK tag antibody beads (Wako) overnight at 4 °C. The resin was washed three times with 1.0 mL of a buffer containing 25 mM Tris-HCl (pH 7.5), 0.15 M NaCl, and 0.01% GDN. Bound proteins were eluted in 20 μL of a buffer containing 0.3 M glycine-HCl (pH 3.5), 0.75 M NaCl, and 0.01% GDN. A 10-μL aliquot of eluate was separated by SDS–PAGE and detected by western blot analysis.

## TOPFlash luciferase reporter assays

HEK293T cells (ATCC, CRL-3216) were grown in DMEM (Nacalai Tesque, 08458-45) complemented with 10% FBS (Thermo Fisher Scientific) and 2 mM L-Glutamine (Nacalai Tesque) at 37 °C in an atmosphere with 5% $CO_2$. The codon-optimized DNA fragment encoding full-length human ZNRF3 (residue 1–936, UniProt accession number Q9ULT6) was synthesized (Thermo Fisher Scientific) and cloned into the pEZT-BM vector with a C-terminal TEV protease cleavage site followed by FLAG and His8 tags. Sequence mismatches were introduced to prevent targeting by the small interfering RNA (siRNA) used in this study. The gene encoding full-length human LGR4 (residue 1–951, UniProt accession number Q9BXB1), carrying the siRNA-resistant mutations that prevent targeting by the siRNA used in this study, was purchased (RIKEN DNA Bank, #HKR247353) and cloned into the pEZT-BM vector with a C-terminal TEV protease cleavage site followed by FLAG and His8 tags. The mutants of ZNRF3 or LGR4 were constructed by site-directed mutagenesis. M50 Super 8x TOPFlash (TOPFlash reporter plasmid) was a gift from Randall Moon (Addgene plasmid # 12456)[50]. L Wnt-3A cells were purchased from ATCC (CRL-2647), and Wnt3a conditioned medium (CM) was prepared according to the product sheet.

TOPFlash luciferase reporter assays were performed as described previously[27] with some modifications. HEK293T cells were added into 96-well collagen-coated plate at $3 \times 10^4$ cells/well, and transfected with siRNA against LGR4 (Applied Biosystems, 4427037, siRNA ID s229315), LGR5 (siRNA ID s16275), ZNRF3 (siRNA ID s38543), or RNF43 (siRNA ID s29699), or control siRNA (Applied Biosystems, 4390843) by reverse transfection method using Lipofectamine RNAiMAX (Invitrogen) according to the manufacturer's protocol. After a medium change following 24 h of incubation, the cells were transiently transfected with 10 ng LGR4 (WT), 20 ng LGR4 (ΔLRR15-LRRCT), 10 ng LGR4 (CD4 TMD), 1 ng ZNRF3 (WT), or 1 ng ZNRF3 (E95N/E97T) expression plasmids, 25 ng of M50 Super 8x TOPFlash, 5 ng of pRL-TK (HSV TK promoter-Renilla Luc, internal control, Promega), as well as pEZT-BM empty vector (control), using Lipofectamine LTX with Plus reagent (Invitrogen) according to the manufacturer's protocol. After incubation for 24 h, the cells were treated with recombinant human RSPO2 (R&D Systems, 3266-RS) and 5% Wnt3a CM, followed by incubation for an additional 24 h. The Firefly and Renilla luciferase activities were measured with Dual-Glo Luciferase assay system (Promega) by GloMax Explorer (Promega). The relative TOPFlash reporter activity was calculated by dividing the TOPFlash reporter activity (relative light unit; RLU) by the Renilla reporter activity (RLU) and normalizing it to control samples. The experiment was repeated three times with triplicates in each experiment. The quantified data are shown as the mean ± standard error of the mean (SEM) with individual data points. Statistical significances were tested by Two-way ANOVA with two-sided Tukey's test using GraphPad Prism 10.2.3.

The expression levels of ZNRF3 or LGR4 proteins were assessed by western blotting. HEK293T cells transfected as in reporter assay were incubated for 24–30 hours, mixed SDS-PAGE sample buffer supplemented with protease inhibitor cocktail (Nacalai Tesque), and then sonicated. The samples were analyzed by western blotting. The bands were visualized using chemiluminescence reagent: Chemi-Lumi One (Nacalai Tesque).

## Cell surface protein isolation

HEK293T cells were seeded on 10-cm collagen-coated dish at $2 \times 10^6$ cells/10 mL medium, transfected with full-length LGR4 (WT or mutant)-FLAG or ZNRF3 (WT or mutant)-FLAG expression plasmids using 10 µg PEI (Polyethylenimine "Max", MW40,000; Polysciences, Inc., USA), and incubated for 24 h. To detect cell surface ZNRF3 proteins, the cells were incubated with 10 µM MG132 (FUJIFILM Wako, 135-18453) for additional 6 h. After incubation at 4 °C for 10 min, the cell surface proteins were biotinylated and pulled down with avidin agarose using Pierce Cell Surface Protein Isolation Kit (Thermo Fisher Scientific, 89881). The cell surface proteins were eluted with SDS sample buffer (62.5 mM Tris–HCl pH 7.5, 1% SDS, 10% glycerol, 50 mM DTT) and analysed by western blotting.

## Reporting summary

Further information on research design is available in the Nature Portfolio Reporting Summary linked to this article.

## Data availability

The Cryo-EM maps and related structure coordinates of the LGR4, LGR4-RSPO2, LGR4-RSPO2-ZNRF3$_{1:1:2}$, and LGR4-RSPO2-ZNRF3$_{2:2:2}$ complexes in this study have been deposited in the EMDB and PDB under accession codes EMD-62218 (PDB 9KB6), EMD-62219 (PDB 9KB7), EMD-62220 (PDB 9KB8), and EMD-62221 (PDB 9KB9), respectively. Previously determined structure coordinates used in this study are PDB 4C8C, 4UFS, 4UFR, 4C9E, 4FII, 7FIJ and 7FII. Source data are provided with this paper.

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

## Acknowledgements

We thank Yoichi Sakamaki and Masahide Kikkawa for their management and support of the Graduate School of Medicine Cryo-EM Facility at the University of Tokyo. This work was supported by a Grant-in-Aid from the Japanese Ministry of Education, Culture, Sports, Science, and Technology, Grant Nos. 22K15046 and 24K09349 (Z.Z.); 22H05184 and 23H00366 (T.S.); and 22H02556 (U.O.). This work was partially supported by the Platform Project for Supporting Drug Discovery and Life Science Research (Basis for Supporting Innovative Drug Discovery and Life Science Research) from AMED (grant number JP21am0101115; support nos. 1570, 1846, and 1848).

## Author contributions

P.Y., A.F., and U.O. designed the experiments. P.Y. and J.A. prepared recombinant proteins. P.Y., J.A., Z.Z., and U.O. performed cryo-EM analyses. P.Y. and A.F. performed the cellular reporter assay. P.Y. and U.O. wrote the original draft with assistance from all the authors. P.Y., A.F., T.S., and U.O. reviewed and edited the manuscript. T.S. and U.O. supervised the project.

## Competing interests

The authors declare no competing interests.
