## [Transparent Peer Review file · Nature Communications]

Structural insights into Wnt/ β -catenin signaling regulation by LGR4, R-spondin, and ZNRF3

Corresponding Author: Dr Umeharu Ohto

Version 0:

Reviewer comments:

Reviewer #1

(Remarks to the Author)

The manuscript by Peng et al. reported multiple structures of LGR4ECD-7TM in complex with RSPO2 and the E3 ligase ZNRF3ECD-TM solved by cryo-EM. These structures provided important insights into the interactions and architecture of the three proteins for the first time. It clearly represents a major step in understanding how LGR4-RSPO2-ZNRF3 regulates Wnt signaling. The authors proposed a model that RSPO2 binds to LGR4 first, and the RSPO2-LGR4 complex binds to a ZNRF3 dimer to inhibit interaction of ZNRF3 with Wnt receptors and induce self-ubiquitination of ZNRF3. However, this model is not totally congruent with recent data about the structures and mechanism of LGR4-RSPO2-ZNRF3 signaling, including that (1) RSPO2 is able to block ZNRF3's ubiquitination of FZD and induce self-ubiquitination of ZNRF3 without any LGR; LGR4 just makes RSPO2 much more potent. (2) both full-length LGR4 and ZNRF3 exist as dimers on cell surface by themselves; (3) full-length LGR4 interacts with ZNRF3 without RSPO which is supported by the structures reported in this manuscript. The key remaining questions is how RSPO binding to ZNRF3 induces self-ubiquitination of the E3 ligase that leads to its own clearance, and how LGR4 enhances this processes. Here are the concerns about the manuscript:

Major concerns:

1. The authors proposed that the 1:1:2 stoichiometry represents an intermediate state in the formation of the fully active 2:2:2 ternary complex. However, no functional data are provided to support this claim. Even if this model is true, it must be very transient. RSPO2 is able to bind to ZNRF3 with very high affinity, similar to that of binding to LGR4, so there is no reason that RSPO2 shall bind to LGR4 first, then to ZNRF3. RSPO1 and RSPO4 has low to no affinity to ZNRF3 by themselves. They authors may test this model using RSPO1 or RSPO4.
2. The significance of the interaction of LGR4ECD with its own 7TM domain should be investigated by mutagenesis. Is this essential for LGR4's structure integrity, or for potentiation of RSPO activity?
3. The interaction of LGR4ECD with ZNRF3 ECD is very intriguing. Is this important for LGR4's function in potentiating RSPO activity? Is this enough for LGR4 to interact with ZNRF3 without RSPO. A pre-existing complex between LGR4 and ZNR3 would explain why RSPO1 and RSPO4 could potentiate Wnt signaling since RSPO1 and RSPO4 have no intrinsic affinity for ZNRF3.
4. The evaluation of ZNRF3 dimerization mutation in Wnt signaling assay needs to be redesigned. HEK293T cells express RN43, ZNRF3, and LGR4 endogenously. The authors knocked down ZNRF3 and transfected with wild-type or mutant, FLAG-tagged ZNRF3. It is hard to see the expression of the recombinant protein based on the Western blot in Figure 4B. The pattern of the bands are similar to that of vector control, just stronger. The band corresponding to recombinant ZNRF3 should be clearly identified. Importantly, the introduction ZNRF3 should decrease Wnt signaling dramatically in the absence of RSPO. Only ~50% reduction was seen. The mutant ZNRF3 also decreased Wnt signaling by ~50%, suggesting the mutation had no effect on its activity in inhibiting Wnt signaling. The mutant did seem to be slightly less sensitive to RSPO2, suggesting that RSPO2 could not induce self-clearance of the mutant. However, this needs be verified by experimental evidence that shows the mutant is resistant to RSPO2-induced self-ubiquitination. Also, knockdown of both RN43 and ZNRF3 are required to test function of ZNRF3 mutant in HEK293T cells.
5. Lastly, a very important question is how RSPO2 induces self-ubiquitination of ZNRF3. Given that ZNRF3 is a dimer, does RSPO2 work by a mechanism similar to that of receptor tyrosine kinases, i.e., ligand binding triggers conformational changes in the enzyme domain to cause cross-ubiquitination? Comparing the structures of full-length ZNRF3 with and without RSPO2 may answer this.

Minor concerns:

1. Page 3, line 56-57, there is no evidence that ever showed that RSPO, LGR5 (full-length), and RNF43/ZNRF3 form a complex to potentiate Wnt signaling.
2. Line 180: "RSPO" should be written as "RSPO2."

Reviewer #2

(Remarks to the Author)

Comments:

The manuscript described results from CryoEM resolving LGR4 alone and in complex with RSPO2 and ZNRF3 pinpointing key interactions within the 2:2:2 complex that are important to understand the RSPO2-mediated regulation of WNT signaling by ubiquitinylation. The study is timely, the findings are novel and the manuscript is well written. The data are well presented and the information is adequately wrapped into figures in the main text as well as the supplementary material.

1. While ZNRF3 acts by ubiquitination of FZDs, it should be underlined that not all ten FZDs might be regulated in a similar way by ZNRF3 (PMID: 38969364).
2. The authors processed the data conventionally with success obtaining 4 structures. Nonetheless, there is either missing information in the workflow or authors could push processing further. Indeed, there is no mention of postprocessing (CTF refinement, polishing) that can improve final maps significantly. You might want to give it a try.
3. Regarding the LGR4/RSPO2/ZNRF32:2:2, as rightfully mentioned TMD quality is pretty bad. Please try to proceed to a local refinement focused on the TMD after particle subtraction on the extracellular section of the complex. This should improve the TMD quality.
4. Reducing the clash score to less than 10 would be good if possible.
5. WNT signaling is not equal WNT/b-catenin signaling. Please define WNT/b-catenin signaling as one branch of the WNT signaling system.
6. The abstract ending "offers insights for therapeutic interventions in Wnt signaling-related disorders" should be formulated more carefully or you should be more informative on which mechanisms can be targeted for therapy.
7. Please avoid the terms WNT ligand and Frizzled receptor. Pharmacologically speaking they do not make sense since WNTs are Frizzled ligands and Frizzleds are WNT receptors.
8. "β-catenin, which translocates to the nucleus and activates the transcription of target genes" – while this is not incorrect, the uninformed reader might think that β-catenin is a transcription factor even though it is a transcriptional regulator of TCF/LEF transcription factors.
9. Line 186 reads "catenine" should be catenin
10. Fig. 1 legend: ribbon should be ribbon
11. Functional validation is scarce in this manuscript. I suggest designing single point mutations disrupting LGR4 ZNRF3 interface and to assess TOPFlash with this system to see if LGR4 potential to potentiate Wnt signaling is impaired
12. Ternary complex is defined by three entities, please define LGR4/RSPO2/ZNRF32:2:2 as heterohexamer or dimer of heterotrimer or diheterotrimer and LGR4/RSPO2/ZNRF31:1:2 as heterotetramer.
13. Fig. 4:
 - a. For the TOPFlash assay: What is 1 n? A technical or biological replicate? Well, I found the answer in the materials and methods but this should be evident from the figure legend. Please provide the individual data points for the independent experiments in a scatter plot. A student's t test is not suitable for the comparison of more than two groups of data. The figure legends in general are short. Some essential information is not provided. In this case, the information about the presence of WNT3A CM is not mentioned and the fact that the HEK293 were pretreated with siRNA against ZNRF3 is not mentioned. Please add this crucial info to the figure legend. It would also be important to understand the level of WNT-induction of the signal. Where is the baseline? How much does WNT3A CM elevate basal? ...and then the emphasis on the RSPO2 effects. Overexpression of ZNRF3 wt and mutant appear to reduce WNT-induced signaling substantially (more than 50 % eyeballing). Given the ZNRF3 downregulation by siRNA and transfection with the mutant ZNRF3 has only a very minor effect on the RSPO2-induced signal amplification – how do the authors explain the relatively small effect of the mutation given that ZNRF3 dimerization is supposed to be very important? How do the authors secure that the expressed siRNA does not target the overexpressed ZNRF3? One recommendation would be to assess the effect of the mutation on the WNT-induced and RSPO-dependent amplification of the signal. For this purpose, it would be most suitable to use the HEK293T RNF43/ZNRF3 double-knockout (R/ZdKO) cell lines (PMID: 38969364).

Regarding the first section of the panel, there is probably something that escapes my understanding, the first condition is not overexpressed for ZNRF3, and ZNRF3 is downregulated by siRNA. Knowing that LGR4/RSPO2 potentiates WNT signaling by scavenging ZNRF3, I would expect no significant difference in activity between conditions with and without RSPO2 when ZNRF3 is silenced. In your experiment, you get a massive shift of activity that suggests that potentiation is not ZNRF3 dependent. How do you explain that?

- b. For the immunoblotting: It appears unusual that the unspecific staining in the absence of ZNRF3 transfection has the same pattern as when the wt and the mutant ZNRF3 are transfected and detected with an anti-FLAG antibody. The authors need to improve this immunoblot to support expression of the ZNRF3 constructs. In fact, the first lane should be practically blank.

Reviewer #3

(Remarks to the Author)

Reviewer #4

(Remarks to the Author)

In this ms, Peng et al., report a cryo-EM structure for the LGR4-RSPO2-ZNRF3 ternary complex. As the authors indicate, there are number of structures for the extracellular domains of the these proteins and their paralogs. However, there is no much information on the contribution of the of LGR and ZNRF3/RNF43 TMDs to these ternary complexes and to their function.

I cannot comment on the structural analyses, but I found the overall model interesting. However, I have a few concerns with the manuscript:

1- although the main thesis for the relevance of these structures is the role of the TMDs, no single mutation experiment was performed to i) validate the importance of possible TMD-TMD interactions in LGR/ZNRF3 function or ii) characterise possible differences between structures obtained without (previous studies e.g. PMID 26123262) and with the TMDs (Their models). The authors only used a previously studied mutant in their reporter assays. What is the take home message for these structures in terms of previously missed protein surface / functional interactions or for the role of TMDs (or the hydrophobic pocket) to promote R-spondin and/or WNT signaling? The authors can address these questions by mutating key residues in LGR, RSPO2, and ZNRF3 with predicted relevance according to their structure, and perform reporter assays with combinations of the 3 proteins. (Please note the general guidelines for mutating and validating residues at the TMD, including ensuring PM trafficking).

2- Please clearly indicate in the model and the results sections that RSPO2 refers to Fu1-Fu2 domains, and not the whole protein.

Version 1:

Reviewer comments:

Reviewer #1

(Remarks to the Author)

The authors have addressed the concerns satisfactorily. No further comment.

Reviewer #2

(Remarks to the Author)

Revision Manuscript 555105:

The authors addressed most of the comments successfully, notably by adding the functional data necessary to validate their structural findings. The additions and revisions improved the manuscript substantially.

However, a few points require clarification or correction:

The authors claim that the clashscore of the structures is less than 10 (see Extended Data Table 1 and rebuttal letter), but the validation reports show substantially different statistics. In particular, two deposited structures have a clashscore of 14. When I requested an updated report for (PDB), it produced a third set of statistics, distinct from both the previous reports and the data table. This raises concerns about consistency between the deposited structures and those used in the structural analysis. Please ensure that all figures, Extended Data Table 1, and structural analyses are updated to match the deposited model.

Line 131: Two consecutive dashes appear in the sentence and seem awkward; they could be removed for smoother reading.

Line 449: Misspelled Frizzled (not frizzeld; and remove the word "receptors" in the same sentence).

Reviewer #3

(Remarks to the Author)

Reviewer #4

(Remarks to the Author)

The authors have addressed my concerns through both functional experiments and more cautious statements of the proposed surface interactions in the complexes.

Response to reviewers

The comments of all reviewers were very useful in helping us to improve the quality of our manuscript. Our responses to the comments of each reviewer can be found below.

The Figures and Table have been reorganized as follows:

Original	Revised
	Fig. 3d (newly created)
	Fig. 3e (newly created)
Fig. 3d	Fig. 3f
Fig. 3e	Fig. 3g
	Fig. 3h (newly created)
	Fig. 4a (newly created)
Fig. 4a	Fig. 4b (modified)
Fig. 4b	Extended Data Fig. 10c,d (modified)
	Fig. 4c (newly created)
	Extended Data Fig. 9b,c (modified)
	Extended Data Fig. 10a,b (newly created)
Extended Data Fig. 10	Extended Data Fig. 11 (moved)

The figure legends have been modified accordingly in the revised manuscript.

Reviewer's comments:

Reviewer #1 (Remarks to the Author)

The manuscript by Peng et al. reported multiple structures of LGR4ECD-7TM in complex with RSPO2 and the E3 ligase ZNRF3ECD-TM solved by cryo-EM. These structures provided important insights into the interactions and architecture of the three proteins for the first time. It clearly represents a major step in understanding how LGR4-RSPO2-ZNRF3 regulates Wnt signaling. The authors proposed a model that RSPO2 binds to LGR4 first, and the RSPO2-LGR4 complex binds to a ZNRF3 dimer to inhibit interaction of ZNRF3 with Wnt receptors and induce self-ubiquitination of ZNRF3. However, this model is not totally congruent with recent data about the structures and mechanism of LGR4-RSPO2-ZNRF3 signaling, including that

Thank you for your positive feedback. We provide point-by-point responses to the specific points you have noted.

(1) RSPO2 is able to block ZNRF3's ubiquitination of FZD and induce self-ubiquitination of ZNRF3 without any LGR

Thank you for your valuable comments. Indeed, a previous study reported that RSPO2 and RSPO3 can enhance Wnt signaling in an LGR-independent manner (Lebensohn, A. M., *elife*, 2018). This does not conflict with our findings, as our study primarily focuses on the LGR4-RSPO-mediated potentiation of Wnt signaling. In the revised manuscript, we have briefly discussed the LGR4-independent mechanism of Wnt signaling regulation in the Discussion section, as follows.

(Page 12, line 238)

“In addition to affecting ZNRF3 ubiquitination activity and promoting its membrane clearance, increasing evidence suggests that LGR4/5 and RSPO may also regulate Wnt/ β -catenin signaling by directly interacting with Wnt receptors, independent of ZNRF3^{36,37,38}. Moreover, RSPO2 and RSPO3 were found to interact with ZNRF3 to potentiate Wnt/ β -catenin signaling even in the absence of LGRs³⁹. However, further research is required to clarify these interactions and their specific contributions to Wnt/ β -catenin signaling.”

(2) both full-length LGR4 and ZNRF3 exist as dimers on cell surface by themselves;

Thank you for pointing this out. We have explained the possible reason why LGR4 was observed as monomers under cryo-EM conditions in the Discussion section. In addition, we have included pull-down data demonstrating that ZNRF3 dimerization is enhanced in the presence of both RSPO and LGR4 (newly created Fig. 4a).

(Page 13, line 251)

“Although full-length LGR4 has been reported to exist as dimers on the cell surface^{26,27}, no dimerized form of LGR4 alone was observed by gel chromatography or cryo-EM analysis, possibly due to its relatively weak dimerization (Extended Data Fig. 5a, b). Moreover, no apparent contact was observed between the two LGR4 protomers within the 2:2:2 heterohexamer. Therefore, it remains unclear whether the preformed LGR4 dimer adopts a configuration distinct from that observed in the heterohexamer.”

(Page 10, line 207)

“We utilized a dimerization-deficient mutant of ZNRF3 (E95N/E97T) (Extended Data Fig. 9a), previously confirmed to be incapable of dimerization²², and tested the dimerization ability of ZNRF3 with or without LGR4 and RSPO2 by pull-down assay (Fig. 4a). The results showed that although ZNRF3 dimerization occurred in the absence of LGR4 and RSPO2, LGR4/RSPO2 co-expression markedly enhanced ZNRF3 dimerization.”

Fig. 4a, Pull-down assay of ZNRNF3 (WT) or ZNRNF3 (E95N/E97T) in the presence or absence of LGR4 and RSPO2. Expi293F cells were transfected with ZNRNF3 ECD-TMD (Flag-tagged), ZNRNF3 ECD-TMD (HA-tagged), LGR4 ECD-TMD (Myc-tagged), and RSPO2 Fu1-2 (Strep-tagged). Cell lysates were incubated with anti-FLAG affinity beads, and bound proteins were detected by western blotting. Data are representative of three independent experiments.

(3) full-length LGR4 interacts with ZNRNF3 without RSPO which is supported by the structures reported in this manuscript.

Thank you for highlighting this. To address the concern, we further validated the interaction between LGR4 and ZNRNF3 using pull-down assay (newly created Fig. 3e). Consistent with the previous reports (Park, S., *Sci Signal.*, 2020), the results showed that LGR4 and ZNRNF3 form a complex even in the absence of RSPO. The amount of LGR4 pulled down by ZNRNF3 was markedly increased in the presence of RSPO2, indicating that RSPO2 stabilizes the ternary complex involving LGR4 and ZNRNF3.

According to the reviewer's suggestion, we revised the sentence as follows:

(Page 9, line 170)

“Notably, this interface appears to facilitate the preassembly of LGR4 and ZNRNF3 into a low-affinity complex at the cell membrane before RSPO engagement (Fig. 3e). RSPO binding then serves as a molecular bridge, stabilizing the interactions and thereby inducing the formation of a more stable ternary complex (Fig. 3e).”

e

Fig. 3e. Pull-down assay of LGR4 and ZNRF3 in the presence or absence of RSPO2. Expi293F cells were transfected with ZNRF3 ECDTMD (Flag-tagged), LGR4 ECD-TMD (Myc-tagged), and RSPO2 Fu1-2 (Strep-tagged). Cell lysates were incubated with anti-FLAG affinity beads, and bound proteins were detected by western blotting. Data are representative of three independent experiments.

Major concerns:

1. The authors proposed that the 1:1:2 stoichiometry represents an intermediate state in the formation of the fully active 2:2:2 ternary complex. However, no functional data are provided to support this claim. Even if this model is true, it must be very transient. RSPO2 is able to bind to ZNRF3 with very high affinity, similar to that of binding to LGR4, so there is no reason that RSPO2 shall bind to LGR4 first, then to ZNRF3. RSPO1 and RSPO4 has low to no affinity to ZNRF3 by themselves. They authors may test this model using RSPO1 or RSPO4.

Thank you for your constructive suggestions. We agree with the reviewer that the sequential binding model is too speculative at this stage. Given that RSPO2 can independently form a complex with ZNRF3 (Park et al. *Sci Signal*), and that LGR4 and ZNRF3 also form a pre-existing, albeit low-affinity, complex (newly created Fig. 3e), it is difficult to determine a definitive binding order. We have revised the Discussion section to reflect this point.

(Page 12, line 225)

“While the interactions between RSPO2 and ZNRF3, as well as between LGR4 and RSPO2, have been well characterized, our study further revealed that LGR4 also establishes direct contact with ZNRF3 (Figs. 3b-e). Each component in the complex engages with the other two, thereby cooperatively stabilizing the overall arrangement of the 2:2:2 heterohexamer and promoting ZNRF3 dimerization required for the potentiation of Wnt/ β -catenin signaling.”

According to the reviewer's suggestion, we reconstituted ternary complexes comprising LGR4, ZNRF3, and either RSPO1 to RSPO4 in Expi293F cells. RSPO1 formed stable LGR4-RSPO1-ZNRF3 complexes (Response Fig. 1) despite its low intrinsic affinity for ZNRF3, although the extent of RSPO1 engagement was lower than that of RSPO2 and RSPO3. RSPO4 failed to form a stable complex with LGR4 and ZNRF3 under these experimental conditions, likely due to its insufficient binding affinity for ZNRF3. These observations align with the previous reports showing that RSPO1 and RSPO4 exhibit lower binding affinity to ZNRF3 compared to RSPO2 and RSPO3.

Response Fig. 1. SDS-PAGE of purified LGR4-RSPO1/2/3/4-ZNRF3 complexes.

2. The significance of the interaction of LGR4ECD with its own 7TM domain should be investigated by mutagenesis. Is this essential for LGR4's structure integrity, or for potentiation of RSPO activity?

Thank you for your insightful suggestion. To investigate the functional significance of the interaction between the LGR4 ECD and TMD domains, we systematically generated multiple single-point mutations at the ECD-TMD interfaces and assessed their impacts on the Wnt signaling using the TOPFlash assay. However, none of these mutations altered signaling activity compared to the wild-type LGR4 (Response Fig. 2). We then deleted the entire connecting segment of LGR4 (Δ aa 526-538) to further examine its role in signaling. This deletion mutant also showed no significant changes in Wnt signaling (Response Fig. 2). Therefore, the functional relevance of the ECD-TMD interface remains unclear at this point.

Response Fig. 2. Relative Luciferase activity in LGR4/LGR5 knockdown HEK293T cells transfected with wild-type LGR4 or ECD-TMD interface mutants of LGR4 and treated with 5% Wnt3a conditioned medium in the presence or absence of 3 ng/mL RSPO2. $n = 2$ independent biological replicates. Bars represent mean and dots show individual data points. EV, empty vector; WT, wild type; CS, connecting segment.

3. The interaction of LGR4ECD with ZNRF3 ECD is very intriguing. Is this important for LGR4's function in potentiating RSPO activity? Is this enough for LGR4 to interact with ZNRF3 without RSPO. A pre-existing complex between LGR4 and ZNRF3 would explain why RSPO1 and RSPO4 could potentiate Wnt signaling since RSPO1 and RSPO4 have no intrinsic affinity for ZNRF3.

Thank you for your insightful comments. We agree that the constitutive interaction between LGR4 ECD and ZNRF3 ECD is critical for understanding the mechanism of RSPO action. Our key findings addressing your points are as follows:

(1) LGR4-ZNRF3 complex pre-exists independently of RSPO2. The pull-down assay confirmed that LGR4 and ZNRF3 form a complex even in the absence of RSPO2 (newly created Fig. 3e). This pre-assembly might explain how RSPO1 and RSPO4, despite their low intrinsic affinity for ZNRF3, can still enhance Wnt signaling.

(2) To test the functional relevance of LGR4-ZNRF3 ECD interaction, we first generated multiple single-point mutants targeting the interface. However, none of these mutants showed reduced reporter activity. This is likely due to the low-affinity nature of the interaction between LGR4 and ZNRF3, making single mutations insufficient to significantly alter the signaling properties. Therefore, we engineered an LGR4 (Δ LRR15–LRRCT) mutant that retains RSPO2 binding region but lacks the ECD region required for association with ZNRF3 ECD. This mutant exhibited >50% reduction in Wnt signaling activity in the TOPFlash assay (newly created Fig. 3d, Extended Data Fig. 10a,b), demonstrating that physical coupling between LGR4 and ZNRF3 ECDs is

important for downstream signaling. We have added these results in the manuscript as follows:
(Page 8, line 163)

“To investigate the functional significance of this interface, we designed a truncated LGR4 variant lacking the ZNRF3-binding region, LRR15-LRRCT domain (Δ aa 390-465), and assessed its activity in Wnt/ β -catenin signaling using the Wnt/ β -catenin reporter TOPFlash system in human embryonic kidney (HEK)293T cells³¹. The LGR4 (Δ LRR15-LRRCT) mutant exhibited significantly reduced Wnt/ β -catenin signaling activity (Fig. 3d, Extended Data Fig. 10b,c), despite being expected to retain RSPO2 binding, suggesting that the newly identified extracellular interface between ZNRF3 and LGR4 is important for Wnt/ β -catenin signaling potentiation. Notably, this interface appears to facilitate the preassembly of LGR4 and ZNRF3 into a low-affinity complex at the cell membrane before RSPO engagement (Fig. 3e). RSPO binding then serves as a molecular bridge, stabilizing the interactions and thereby inducing the formation of a more stable ternary complex (Fig. 3e).”

Fig. 3d. TOPFlash reporter assays using full-length wild-type (WT) or mutant (Δ LRR15-LRRCT) LGR4. HEK293T cells were transfected with siRNA against LGR4 and LGR5, followed by plasmid transfection, and then treated with 5% Wnt3a-conditioned medium in the presence or absence of 3 ng/mL RSPO2. $n = 3$ biological replicates. Bars represent mean \pm standard error of the mean (SEM), and dots show individual data points. Statistical significance was determined using two-way ANOVA with two-sided Tukey’s test. * $P = 0.038$, *** $P < 0.001$. ns, not significant.

Extended Data Fig. 10

a, Western blot analysis of full-length wild-type (WT) LGR4, LGR4 (CD4 TMD) mutant, or LGR4 (Δ LRR15-LRRCT) mutant. LGR4 and LGR5 knockdown HEK293T cells were transfected with empty vector (EV) or FLAG-tagged LGR4 expression plasmids as in TOPFlash reporter assay, and the total cell lysates (TCL) were analyzed in western blotting with anti-FLAG or anti- β actin antibodies.

b, Western blot analysis of cell surface LGR4 (WT), LGR4 (CD4 TMD), or LGR4 (Δ LRR15-LRRCT). HEK293T cells were transfected with FLAG-tagged LGR4 expression plasmids and subjected to membrane biotinylation, and biotinylated membrane proteins were isolated with avidin beads and analyzed by western blotting with anti-FLAG antibodies.

4. The evaluation of ZNRF3 dimerization mutation in Wnt signaling assay needs to be redesigned. HEK293T cells express RNF43, ZNRF3, and LGR4 endogenously. The authors knocked down ZNRF3 and transfected with wild-type or mutant, FLAG-tagged ZNRF3. It is hard to see the expression of the recombinant protein based on the Western blot in Figure 4B. The pattern of the bands are similar to that of vector control, just stronger. The band corresponding to recombinant ZNRF3 should be clearly identified. Importantly, the introduction ZNRF3 should decrease Wnt signaling dramatically in the absence of RSPO. Only ~50% reduction was seen. The mutant ZNRF3 also decreased Wnt signaling by ~50%, suggesting the mutation had no effect on its activity in inhibiting Wnt signaling. The mutant did seem to be slightly less sensitive to RSPO2, suggesting that RSPO2 could not induce self-clearance of the mutant. However, this needs be verified by experimental evidence that shows the mutant is resistant to RSPO2-induced self-ubiquitination. Also, knockdown of both RNF43 and ZNRF3 are required to test function of ZNRF3 mutant in HEK293T cells.

We appreciate your insightful comments and constructive suggestions. In this revision, we have modified the experimental design to obtain more robust and compelling data.

In the original manuscript, we knocked down only ZNRF3 and introduced either wild-type or E95N/E97T ZNRF3 in the TOPFlash reporter assay. Under these conditions, residual endogenous

RNF43 may have masked the effects of exogenously expressed ZNRF3. To address this, we conducted simultaneous knockdown of both RNF43 and ZNRF3 (newly created Fig. 4b). Additionally, we have included data from control siRNA-treated cells and samples without Wnt3A stimulation to strengthen the conclusion.

In cells treated with Wnt3A alone, introduction of either wild-type or E95N/E97T ZNRF3 resulted in a decrease in TOPFlash activity to approximately 20%, indicating that the mutation does not impair the basal inhibitory ZNRF3 function of ZNRF3. In contrast, RSPO2-induced inhibition of ZNRF3 activity was more pronounced in the E95N/E97T mutant than in the wild-type. This result suggests that ZNRF3 dimerization is critical for RSPO2-mediated potentiation of Wnt signaling, possibly by promoting the self-clearance of ZNRF3 from the plasma membrane. Since ZNRF3 dimerization is enhanced in the presence of LGR4 and RSPO2 (newly created Fig. 4a, Zebisch. M., *Nat. Commun.*, 2013), our findings support the functional importance of the LGR4–RSPO2–ZNRF3 complex in regulating ZNRF3 activity.

Furthermore, according to the reviewer’s suggestion, we attempted to experimentally determine whether the mutant ZNRF3 is resistant to RSPO2-induced self-ubiquitination. However, this experiment proved technically challenging. Total ZNRF3 protein levels were markedly increased when co-expressed with LGR4 and RSPO2 (Response Fig. 3), making it difficult to quantitatively compare ubiquitination levels between wild-type and mutant ZNRF3. Indeed, few studies have investigated the ubiquitination of ZNRF3, and to our knowledge, no reports have directly demonstrated RSPO2- and LGR4-mediated enhancement of ZNRF3 self-ubiquitination.

Response Fig. 3. Western blot analysis of ZNRF3 in total cell lysates (TCL) and cell surface fractions (Avidin pull-down), with or without LGR4 expression and RSPO2 treatment. HEK293T cells were transfected with full-length FLAG-tagged ZNRF3 (WT), HA-tagged ubiquitin, with or without Myc-tagged LGR4 expression plasmids. After 24 hours incubation, the cells treated with or without 100 ng/mL RSPO2 for 24 hours. To inhibit proteasomal degradation, 10 μ M MG132 was added during the final 6 hours of incubation. The cells were subjected to membrane

biotinylation, and biotinylated surface proteins were isolated using avidin-conjugated beads.

As shown in the newly presented Fig. 4a, co-expression of LGR4 and RSPO2 promoted dimer formation of wild-type ZNRF3, but not of the mutant. As previously reported (Siepe, D. H., *ACS Synth. Biol.*, 2023), enhanced ZNRF3 dimerization facilitates its clearance from the plasma membrane. These findings suggest that RSPO2 and LGR4 may promote ZNRF3 membrane clearance through a similar mechanism; however, the precise molecular details remain to be elucidated.

Regarding the expression levels of recombinant ZNRF3, we confirmed that the expression levels of ZNRF3(WT) and ZNRF3(E95N/E97T) were equivalent (newly created Extended Data Fig. 10c). The ZNRF3(E95N/E97T) appeared as a higher molecular weight band, suggesting that the mutant was glycosylated, as expected. In addition to immunoblotting of total cell lysates (TCL), we also assessed the presence of ZNRF3 on the plasma membrane (newly created Extended Data Fig. 10d).

Fig. 4. ZNRF3 dimerization through complexation with LGR4 and RSPO2 is important for Wnt signal potentiation.

b, TOPFlash reporter assays of full-length wild-type (WT) or dimerization-deficient mutant (E95N/E97T) ZNRF3. HEK293T cells were transfected with indicated siRNA followed by plasmids transfection, and then treated with or without 5% Wnt3a conditioned medium in the presence or absence of 200 ng/mL RSPO2. $n = 3$ biological replicates. Bars are mean \pm standard error of the mean (SEM) and dots show individual data points. Statistical significance was determined using Two-way ANOVA with two-sided Tukey's test. *** $P < 0.001$. ns, not significant. **c,** Western blotting analysis showed that ZNRF3 (WT) and ZNRF3 (E95N/E97T) exhibited similar expression levels in the TCL. TCL, total cell lysate; EV, empty vector; WT, wild type.

d, ZNRF3 (E95N/E97T) showed reduced ubiquitylation (Ub) levels compared to WT in the absence of LGR4 and RSPO2.

Extended Fig. 10 Expression and cell surface localization of LGR4 and ZNRF3

c, Western blotting analysis of full-length that ZNRF3 (WT) or ZNRF3 (E95N/E97T). ZNRF3 and RNF43 knockdown HEK293T cells were transfected with empty vector (EV) or FLAG-tagged ZNRF3 expression plasmids, and the total cell lysates were analyzed as in **a**.

d, Western blot analysis of cell surface ZNRF3 (WT) or ZNRF3 (E95N/E97T). HEK293T cells were transfected with FLAG-tagged ZNRF3 expression plasmids, treated with 10 μ M MG132 for 6 h to prevent proteasomal degradation, and subjected to membrane biotinylation. The biotinylated membrane proteins were isolated with avidin beads and analyzed by western blotting with anti-FLAG antibodies.

5. Lastly, a very important question is how RSPO2 induces self-ubiquitination of ZNRF3. Given that ZNRF3 is a dimer, does RSPO2 work by a mechanism similar to that of receptor tyrosine kinases, i.e., ligand binding triggers conformational changes in the enzyme domain to cause cross-ubiquitination? Comparing the structures of full-length ZNRF3 with and without RSPO2 may answer this.

We sincerely appreciate your insightful suggestions regarding the mechanism by which RSPO2 induces ZNRF3 self-ubiquitination. Although we successfully prepared the ZNRF3–RSPO2 complex by co-transfection in Expi293F cells, we encountered challenges in obtaining a high-resolution cryo-EM reconstruction, possibly due to the inherent flexibility of the single transmembrane helices of ZNRF3 and the relatively small molecular size of the complex for cryo-EM analysis. Although our revised data show that LGR4-RSPO2 binding robustly enhances ZNRF3 dimerization (newly created Fig. 4a), consistent with the previous observations (Zebisch, M., *Nat Commun.*, 2013), the precise mechanism remains unclear and needs further investigation. Considering these findings, we revised the Discussion section:

(Page13, line 245)

“The mechanism by which LGR4- and RSPO-induced ZNRF3 dimerization potentiates Wnt/ β -catenin signaling has not been fully elucidated. Receptor tyrosine kinases undergo ligand-induced

dimerization, triggering trans-autophosphorylation and subsequent activation^{40,41}. Similarly, ZNRF3 dimerization may facilitate its self-ubiquitination and regulatory function, thereby enhancing the Wnt/ β -catenin signaling. Future structural investigations of ligand-induced conformational changes in full-length ZNRF3 may clarify the underlying mechanism.”

Fig. 4a, Pull-down assay showing dimerization of ZNRF3 (WT) and ZNRF3 (E95N/E97T) in the presence or absence of LGR4 and RSPO2. Expi293F cells were transfected with ZNRF3 ECNTMD (Flag-tagged), ZNRF3 ECNTMD (HA-tagged), LGR4 ECNTMD (Myc-tagged), and RSPO2 Fu1-2 (Strep-tagged). Cell lysates were subjected to anti-FLAG affinity beads, and bounded proteins were detected by western blotting. Data are representative of three independent experiments.

Minor concerns:

1. Page 3, line 56-57, there is no evidence that ever showed that RSPO, LGR5 (full-length), and RNF43/ZNRF3 form a complex to potentiate Wnt signaling.

Thank you for pointing this out. We have removed the sentence referring to LGR5 in the revised manuscript.

(Page 3, line 62)

2. Line 180: "RSPO" should be written as "RSPO2."

Thank you for pointing this out. We have revised the manuscript from “RSPO” to “RSPO2”

(Page 10, line 203).

Reviewer #2 (Remarks to the Author):

Comments:

The manuscript described results from CryoEM resolving LGR4 alone and in complex with RSPO2 and ZNRF3 pinpointing key interactions within the 2:2:2 complex that are important to understand the RSPO2-mediated regulation of WNT signaling by ubiquitylation. The study is timely, the findings are novel and the manuscript is well written. The data are well presented and the information is adequately wrapped into figures in the main text as well as the supplementary material.

1. While ZNRF3 acts by ubiquitination of FZDs, it should be underlined that not all ten FZDs might be regulated in a similar way by ZNRF3 (PMID: 38969364).

Thank you for pointing this out. We have replaced the term “all FZDs” with “a subset of FZDs” in the revised manuscript and have added the appropriate references.

(Page 3, line 56)

2. The authors processed the data conventionally with success obtaining 4 structures. Nonetheless, there is either missing information in the workflow or authors could push processing further. Indeed, there is no mention of postprocessing (CTF refinement, polishing) that can improve final maps significantly. You might want to give it a try.

We apologize for the omission of some procedures in the cryo-EM workflow in the original manuscript. In fact, CTF refinement was consistently applied during NU refinement. Reference based motion correction implemented in CryoSPARC was also utilized; however, only LGR4 alone dataset showed a significant improvement in map quality. The updated cryo-EM workflows are now provided in Extended Data Figs. 5, 6, and 8.

3. Regarding the LGR4/RSPO2/ZNRF3_{2:2:2}, as rightfully mentioned TMD quality is pretty bad. Please try to proceed to a local refinement focused on the TMD after particle subtraction on the extracellular section of the complex. This should improve the TMD quality.

Thank you for your constructive suggestion. We have already attempted focused refinement on the TMD region to improve its map quality in the 2:2:2 complex. However, no improvement was observed; instead, the resulting map showed reduced quality, possibly due to the inherent flexibility of one TMD protomer relative to the other.

4. Reducing the clash score to less than 10 would be good if possible.

Thank you for pointing this out. We further refined the model of the LGR4–RSPO2–ZNRF3_{2:2:2} complex, resulting in a reduced clash score of less than 10.

Response Table. 1. Summary of the clash score and geometry metrics for the new LGR4–RSPO2–ZNRF3_{2:2:2} complex.

	Clash score (new)	Ramachandran outliers (New)	Bonds length (Å) (New)	Bonds Angle (°) (New)
LGR4-RSPO2-ZNRF3 _{2:2:2} heterohexamer	9.69	2.46	0.003	0.655

5. WNT signaling is not equal WNT/b-catenin signaling. Please define WNT/b-catenin signaling as one branch of the WNT signaling system.

Thank you for helping us clarify this concept. We have replaced the term “Wnt signaling” with “Wnt/β-catenin signaling” throughout the revised manuscript.

6. The abstract ending “offers insights for therapeutic interventions in Wnt signaling-related disorders” should be formulated more carefully or you should be more informative on which mechanisms can be targeted for therapy.

Thank you for your comment. To avoid ambiguity, we have revised the manuscript as follows:
(Page 2, line 35)

“This study provides a structural basis for understanding the regulatory mechanism of Wnt/β-catenin signaling through the LGR4-RSPO2-ZNRF3 pathway and may offer opportunities for future drug development targeting this axis.”

7. Please avoid the terms WNT ligand and Frizzled receptor. Pharmacologically speaking they do not make sense since WNTs are Frizzled ligands and Frizzleds are WNT receptors.

Thank you for helping us clarify this concept. we have revised the manuscript as follows:
(Page 3, line 45)

“Extracellular Wnt proteins bind to the seven-transmembrane receptors Frizzleds (FZDs) on the cell surface, in conjunction with single-transmembrane low-density lipoprotein receptor-related protein (LRP) 5/6, to initiate the Wnt/β-catenin pathway.”

8. “β-catenin, which translocates to the nucleus and activates the transcription of target genes” – while this is not incorrect, the uninformed reader might think that b-catenin is a transcription factor even though it is a transcriptional regulator of TCF/LEF transcription factors.

Thank you for pointing this out. We have made appropriate modifications to the original text as follows.

(Page 3, line 48)

“This interaction promotes the accumulation of cytosolic β-catenin, which then translocates to the

nucleus and activates TCF/LEF transcription factors, triggering the expression of target genes^{6,7}”

9. Lane 186 reads “catenine” should be catenin

Thank you for pointing this out. We have corrected this mistake.
(Page 10, line 215).

10. Fig. 1 legend: qribbon should be ribbon

Thank you for pointing this out. We have corrected this mistake.
(Page 20, line 382)

11. Functional validation is scarce in this manuscript. I suggest designing single point mutations disrupting LGR4 ZNRF3 interface and to asses TOPFlash with this system to see if LGR4 potential to potentiate Wnt signaling is impaired

Thank you for your constructive comment. In response to the reviewer’s suggestion, we generated multiple single-point mutants targeting the interface. However, none of these mutants showed reduced reporter activity. This is likely due to the low-affinity nature of the interaction between LGR4 and ZNRF3, making single mutations insufficient to significantly alter the signaling properties.

Therefore, to test the functional relevance of LGR4-ZNRF3 ECD interaction, we engineered an LGR4 (Δ LRR15–LRRCT) mutant that retains RSPO2 binding region but lacks the ECD region required for association with ZNRF3 ECD. This mutant exhibited >50% reduction in Wnt signaling activity in the TOPFlash assay (newly created Fig. 3d, Extended Data Fig. 10a,b), demonstrating that physical coupling between LGR4 and ZNRF3 ECDs is important for downstream signaling. We have added these results in the manuscript as follows:

(Page 8, line 163)

“To investigate the functional significance of this interface, we designed a truncated LGR4 variant lacking the ZNRF3-binding region, LRR15-LRRCT domain (Δ aa 390-465), and assessed its activity in Wnt/ β -catenin signaling using the Wnt/ β -catenin reporter TOPFlash system in human embryonic kidney (HEK)293T cells³¹. The LGR4 (Δ LRR15-LRRCT) mutant exhibited significantly reduced Wnt/ β -catenin signaling activity (Fig. 3d, Extended Data Fig. 10b,c), despite being expected to retain RSPO2 binding, suggesting that the newly identified extracellular interface between ZNRF3 and LGR4 is important for Wnt/ β -catenin signaling potentiation. Notably, this interface appears to facilitate the preassembly of LGR4 and ZNRF3 into a low-affinity complex at the cell membrane before RSPO engagement (Fig. 3e). RSPO binding then serves as a molecular bridge, stabilizing the interactions and thereby inducing the formation of a more stable ternary complex (Fig. 3e).”

Fig. 3d. TOPFlash reporter assays using full-length wild-type (WT) or mutant (Δ LRR15-LRRCT) LGR4. HEK293T cells were transfected with siRNA against LGR4 and LGR5, followed by plasmid transfection, and then treated with 5% Wnt3a-conditioned medium in the presence or absence of 3 ng/mL RSPO2. $n = 3$ biological replicates. Bars represent mean \pm standard error of the mean (SEM), and dots show individual data points. Statistical significance was determined using two-way ANOVA with two-sided Tukey's test. * $P = 0.038$, *** $P < 0.001$. ns, not significant.

Extended Data Fig. 10

a, Western blot analysis of full-length wild-type (WT) LGR4, LGR4 (CD4 TMD) mutant, or LGR4 (Δ LRR15-LRRCT) mutant. LGR4 and LGR5 knockdown HEK293T cells were transfected with empty vector (EV) or FLAG-tagged LGR4 expression plasmids as in TOPFlash reporter assay, and the total cell lysates (TCL) were analyzed in western blotting with anti-FLAG or anti- β actin antibodies.

b, Western blot analysis of cell surface LGR4 (WT), LGR4 (CD4 TMD), or LGR4 (Δ LRR15-LRRCT). HEK293T cells were transfected with FLAG-tagged LGR4 expression plasmids and subjected to membrane biotinylation, and biotinylated membrane proteins were isolated with avidin beads and analyzed by western blotting with anti-FLAG antibodies.

12. Ternary complex is defined by three entities, please define LGR4/RSPO2/ZNRF3_{2:2:2} as heterohexamer or dimer of heterotrimer or diheterotrimer and LGR4/RSPO2/ZNRF3_{1:1:2} as heterotetramer.

Thank you for your suggestions. We have defined the LGR4/RSPO2/ZNRF3_{2:2:2} complex as heterohexamer and the LGR4/RSPO2/ZNRF3_{1:1:2} complex as heterotetramer accordingly.

(Page 7, line 132)

13. Fig. 4:

a. For the TOPFlash assay: What is 1 n? A technical or biological replicate? Well, I found the answer in the materials and methods but this should be evident from the figure legend. Please provide the individual data points for the independent experiments in a scatter plot. A student's t test is not suitable for the comparison of more than two groups of data. The figure legends in general are short. Some essential information is not provided. In this case, the information about the presence of WNT3A CM is not mentioned and the fact that the HEK293 were pretreated with siRNA against ZNRF3 is not mentioned. Please add this crucial info to the figure legend. It would also be important to understand the level of WNT-induction of the signal. Where is the baseline? How much does WNT3A CM elevate basal? ...and then the emphasis on the RSPO2 effects. Overexpression of ZNRF3 wt and mutant appear to reduce WNT-induced signaling substantially (more than 50 % eyeballing).

Given the ZNRF3 downregulation by siRNA and transfection with the mutant ZNRF3 has only a very minor effect on the RSPO2-induced signal amplification – how do the authors explain the relatively small effect of the mutation given that ZNRF3 dimerization is supposed to be very important?

How do the authors secure that the expressed siRNA does not target the overexpressed ZNRF3? One recommendation would be to assess the effect of the mutation on the WNT-induced and RSPO-dependent amplification of the signal. For this purpose, it would be most suitable to use the HEK293T RNF43/ZNRF3 double-knockout (R/ZdKO) cell lines (PMID: 38969364).

Regarding the first section of the panel, there is probably something that escapes my understanding, the first condition is not overexpressed for ZNRF3, and ZNRF3 is downregulated by siRNA. Knowing that LGR4/RSPO2 potentiates WNT signaling by scavenging ZNRF3, I would expect no significant difference in activity between conditions with and without RSPO2 when ZNRF3 is silenced. In your experiment, you get a massive shift of activity that suggests that potentiation is not ZNRF3 dependent. How do you explain that?

b. For the immunoblotting: It appears unusual that the unspecific staining in the absence of ZNRF3 transfection has the same pattern as when the wt and the mutant ZNRF3 are transfected and detected with an anti-FLAG antibody. The authors need to improve this immunoblot to

support expression of the ZNRF3 constructs. In fact, the first lane should be practically blank.

We appreciate your insightful comments and constructive suggestions. In this revision, we have modified the experimental design, which enabled us to obtain more robust and compelling data.

The ZNRF3 expression plasmid contains a codon-optimized sequence for expression in human cells and includes four nucleotide mismatches within the siRNA target site, thereby preventing knockdown of the overexpressed ZNRF3. To avoid confusion, this detail has been clarified in “TOPFlash luciferase reporter assay” in Methods section as “The gene encoding full-length human ZNRF3 (residue 1-936, UniPort accession number Q9ULT6), which was codon-optimized for expression in human cells and contained sequence mismatches that prevent targeting by the siRNA used in this study, ...” (Page 27, line 565).

In the original manuscript, we knocked down only ZNRF3 and introduced either wild-type or E95N/E97T ZNRF3 in the TOPFlash reporter assay. Under these conditions, residual endogenous RNF43 may have masked the effects of exogenously expressed ZNRF3. To address this, we conducted simultaneous knockdown of both RNF43 and ZNRF3 (newly created Fig. 4b). Additionally, we have included data from control siRNA-treated cells and samples without Wnt3A stimulation to strengthen the conclusion. The statistical significance was determined using Two-way ANOVA with two-sided Tukey’s test. In cells with both ZNRF3 and RNF43 knocked down, reporter activity did not increase substantially upon treatment with both Wnt3a and RSPO2, compared to treatment with Wnt3a alone (Fig. 3b, gray bar). In cells treated with Wnt3A alone, introduction of wild-type or E95N/E97T ZNRF3 decreased TOPFlash activity at the comparable level, indicating that the mutation does not impair ZNRF3’s inhibitory function. In contrast, RSPO2-induced inhibition of ZNRF3 activity was more pronounced in the E95N/E97T mutant than in the wild-type. This result suggests that ZNRF3 dimerization is critical for RSPO2-mediated self-clearance. Since ZNRF3 dimerization is enhanced in the presence of LGR4 and RSPO2 (newly created Fig. 4a, Zebisch. M., *Nat. Commun.*, 2013), our findings support the functional importance of the LGR4–RSPO2–ZNRF3 complex in regulating ZNRF3 activity.

Regarding the expression levels of recombinant ZNRF3, we confirmed that the expression levels of ZNRF3(WT) and ZNRF3(E95N/E97T) were equivalent (newly created Extended Data Fig. 10c). The ZNRF3(E95N/E97T) appeared as a higher molecular weight band, suggesting that the mutant was glycosylated, as expected. In addition to immunoblotting of total cell lysates (TCL), we also assessed the presence of ZNRF3 on the plasma membrane (newly created Extended Data Fig. 10d).

Regarding the figure and its legend, we have added individual data points to the graph (newly created Fig. 4b) and included additional experimental and statistical details in the legend, as follows:

Fig. 4b, TOPFlash reporter assays using full-length wild-type (WT) or dimerization-deficient mutant (E95N/E97T) ZNR3. HEK293T cells were transfected with the indicated siRNA, followed by plasmid transfection, and then treated with or without 5% Wnt3a-conditioned medium in the presence or absence of 200 ng/mL RSPO2. $n = 3$ biological replicates. Bars represent mean \pm standard error of the mean (SEM), and dots show individual data points. Statistical significance was determined using Two-way ANOVA with two-sided Tukey's test. *** $P < 0.001$. ns, not significant.

Extended Fig. 10 Expression and cell surface localization of LGR4 and ZNR3

c, Western blotting analysis of full-length that ZNR3 (WT) or ZNR3 (E95N/E97T). ZNR3 and RNF43 knockdown HEK293T cells were transfected with empty vector (EV) or FLAG-tagged ZNR3 expression plasmids, and the total cell lysates were analyzed as in **a**.

d, Western blot analysis of cell surface ZNR3 (WT) or ZNR3 (E95N/E97T). HEK293T cells were transfected with FLAG-tagged ZNR3 expression plasmids, treated with 10 μ M MG132 for 6 h to prevent proteasomal degradation, and subjected to membrane biotinylation. The biotinylated membrane proteins were isolated with avidin beads and analyzed by western blotting with anti-

FLAG antibodies.

Reviewer #3 (Remarks to the Author):

We would like to thank the reviewer for his/her valuable time in evaluating our manuscript.

Reviewer #4 (Remarks to the Author)

In this ms, Peng et al., report a cryo-EM structure for the LGR4-RSPO2-ZNRF3 ternary complex. As the authors indicate, there are number of structures for the extracellular domains of the these proteins and their paralogs. However, there is no much information on the contribution of the of LGR and ZNRF3/RNF43 TMDs to these ternary complexes and to their function.

I cannot comment on the structural analyses, but I found the overall model interesting. However, I have a few concerns with the manuscript:

1- although the main thesis for the relevance of these structures is the role of the TMDs, no single mutation experiment was performed to

i) validate the importance of possible TMD-TMD interactions in LGR/ZNRF3 function or

Thank you for your valuable suggestions. Due to the limited map quality of the TMD region in our structure, the interaction between the TMDs of LGR4 and ZNRF3 could not be clearly resolved. Therefore, validation by point mutation appears to be challenging. To address this issue, we generated a chimeric LGR4 mutant by replacing the TMD of LGR4 with the single transmembrane helix of human CD4, based on a previous study (Carmon, K. S., *Mol Cell Biol*, 2012). The LGR4 (CD4 TMD) mutant exhibited a significantly impaired ability to enhance Wnt signaling, likely due to compromised interaction with the ZNRF3 TMD (newly created Fig. 3h, Extended Data Fig. 10b,c). We have added this finding in the manuscript as follows:

(Page 9, line 191)

“When the TMD of LGR4 (7-TM) was replaced with that of CD4 (1-TM) according to the previous study²⁹, the resulting LGR4 variant (CD4 TMD) showed nearly a 50% reduction in RSPO2-induced Wnt/ β -catenin signaling activity compared to the wild-type LGR4 (Fig. 3h, Extended Data Fig. 10a,b). These results suggest that the spatial constraints of the TM helices of ZNRF3 imposed by the 7-TM TMD of LGR4 may be important for the potentiation of Wnt/ β -catenin signaling.”

Fig. 3h. TOPFlash reporter assays using full-length wild-type (WT) or mutant (CD4 TMD) LGR4. HEK293T cells were stimulated as in **d**. $n = 3$ biological replicates. Bars represent mean \pm SEM, and dots show individual data points. Statistical significance was determined using two-way ANOVA with two-sided Tukey's test. *** $P < 0.001$. ns, not significant.

Extended Data Fig. 10

a, Western blot analysis of full-length wild-type (WT) LGR4, LGR4 (CD4 TMD) mutant, or LGR4 (Δ LRR15-LRRCT) mutant. LGR4 and LGR5 knockdown HEK293T cells were transfected with empty vector (EV) or FLAG-tagged LGR4 expression plasmids as in TOPFlash reporter assay, and the total cell lysates (TCL) were analyzed in western blotting with anti-FLAG or anti- β actin antibodies.

b, Western blot analysis of cell surface LGR4 (WT), LGR4 (CD4 TMD), or LGR4 (Δ LRR15-LRRCT). HEK293T cells were transfected with FLAG-tagged LGR4 expression plasmids and subjected to membrane biotinylation, and biotinylated membrane proteins were isolated with avidin beads and analyzed by western blotting with anti-FLAG antibodies.

ii) characterise possible differences between structures obtained without (previous studies e.g. PMID 26123262) and with the TMDs (Their models).

Thank you for your valuable suggestions. We have made structural comparisons between the LGR4-RSPO2 complex, the LGR4-RSPO2-ZNRF3 complex in this study, as well as the

previously reported LGR5-RSPO2-ZNRF3 complexes (Extended Data Fig. 9b,c, Zebisch, M., *J Struct Biol.* 2015) and discussed this in the Results section (Page 7, line 145).

“The 2:2:2 complex was generally consistent with the previously predicted 2:2:2 model²⁶, created by superposing two LGR4-RSPO2 complexes onto each RSPO2 protomer of the ZNRF3-RSPO2 complex, as well as the low-resolution crystal structure of the LGR5_{ECD}-RSPO2-ZNRF3_{ECD} 2:2:2 complex²⁵ (Extended Data Fig. 9b).”

Extended Data Fig. 9.

b, Superimposition of the cryo-EM structure of the LGR4-RSPO2-ZNRF3_{2:2:2} complex (this study) with that of the LGR5-RSPO2-ZNRF3 complex (PDB: 4UFS).

c, Superimposition of the cryo-EM structure of the LGR4-RSPO2 complex (this study), the LGR4-RSPO2-ZNRF3_{2:2:2} complex (this study) with the crystal structure of the LGR5-RSPO2 complex (PDB: 4UFR).

The authors only used a previously studied mutant in their reporter assays. What is the take home message for these structures in terms of previously missed protein surface / functional interactions or for the role of TMDs (or the hydrophobic pocket) to promote R-spondin and/or WNT signaling? The authors can address these questions by mutating key residues in LGR, RSPO2, and ZNRF3 with predicted relevance according to their structure, and perform reporter assays with combinations of the 3 proteins. (Please note the general guidelines for mutating and validating residues at the TMD, including ensuring PM trafficking).

Thank you for your valuable suggestions. In this revision, we have added three newly conducted functional experiments (Fig. 3d, 3h, and Fig. 4b), which demonstrate the following:

Fig. 3d: the importance of the newly identified LGR4-ZNRF3 ECD-ECD interface

Fig. 3h: the importance of the newly identified LGR4-ZNRF3 TMD-TMD interface

Fig. 4b: the importance of ZNRF3 dimerization

To specifically test the functional relevance of the newly identified LGR4-ZNRF3 ECD-ECD interaction, we engineered an LGR4 (Δ LRR15-LRRCT) mutant that retains RSPO2 binding

region but lacks the ECD region required for association with ZNRF3 ECD. This mutant exhibited >50% reduction in Wnt signaling activity in the TOPFlash assay (newly created Fig. 3d, Extended Data Fig. 10a,b), demonstrating that physical coupling between LGR4 and ZNRF3 ECDs is important for downstream signaling. We have added these results in the manuscript as follows: (Page 8, line 163)

“To investigate the functional significance of this interface, we designed a truncated LGR4 variant lacking the ZNRF3-binding region, LRR15-LRRCT domain (Δ aa 390-465), and assessed its activity in Wnt/ β -catenin signaling using the Wnt/ β -catenin reporter TOPFlash system in human embryonic kidney (HEK)293T cells³¹. The LGR4 (Δ LRR15-LRRCT) mutant exhibited significantly reduced Wnt/ β -catenin signaling activity (Fig. 3d, Extended Data Fig. 10b,c), despite being expected to retain RSPO2 binding, suggesting that the newly identified extracellular interface between ZNRF3 and LGR4 is important for Wnt/ β -catenin signaling potentiation. Notably, this interface appears to facilitate the preassembly of LGR4 and ZNRF3 into a low-affinity complex at the cell membrane before RSPO engagement (Fig. 3e). RSPO binding then serves as a molecular bridge, stabilizing the interactions and thereby inducing the formation of a more stable ternary complex (Fig. 3e).”

To ensure that the LGR4 mutant was correctly expressed on the plasma membrane, we performed Cell-surface Biotinylation Assay and compared the membrane expression levels of wild-type and mutant LGR4. The results of western blot analysis showed that the wild type and the mutant LGR4 were expressed at comparable levels on the cell membrane. (Extended Data Fig. 10a,b).

Fig. 3d. TOPFlash reporter assays using full-length wild-type (WT) or mutant (Δ LRR15-LRRCT) LGR4. HEK293T cells were transfected with siRNA against LGR4 and LGR5, followed by plasmid transfection, and then treated with 5% Wnt3a-conditioned medium in the presence or absence of 3 ng/mL RSPO2. $n = 3$ biological replicates. Bars represent mean \pm standard error of the mean (SEM), and dots show individual data points. Statistical significance was determined

using two-way ANOVA with two-sided Tukey's test. * $P = 0.038$, *** $P < 0.001$. ns, not significant.

Extended Data Fig. 10

a, Western blot analysis of full-length wild-type (WT) LGR4, LGR4 (CD4 TMD) mutant, or LGR4 (Δ LRR15-LRRCT) mutant. LGR4 and LGR5 knockdown HEK293T cells were transfected with empty vector (EV) or FLAG-tagged LGR4 expression plasmids as in TOPFlash reporter assay, and the total cell lysates (TCL) were analyzed in western blotting with anti-FLAG or anti- β actin antibodies.

b, Western blot analysis of cell surface LGR4 (WT), LGR4 (CD4 TMD), or LGR4 (Δ LRR15-LRRCT). HEK293T cells were transfected with FLAG-tagged LGR4 expression plasmids and subjected to membrane biotinylation, and biotinylated membrane proteins were isolated with avidin beads and analyzed by western blotting with anti-FLAG antibodies.

2- Please clearly indicate in the model and the results sections that RSPO2 refers to Fu1-Fu2 domains, and not the whole protein.

Thank you for your valuable suggestions. As suggested, we have revised the manuscript to specify “RSPO2 (residues 20-143, Fu1-2 domains)” (Page 6, line 110) and updated the corresponding figure to clearly indicate that RSPO2 corresponds to the Fu1-2 region (Extended Data Fig. 7a).

Extended Data Fig. 7a. The structure of RSPO2 Fu1-2 domains (this study), consisting of 6 pairs of β -hairpins stabilized by disulfide bonds (yellow).

Response to reviewers

Reviewer's comments:

Reviewer #1 (Remarks to the Author):

The authors have addressed the concerns satisfactorily. No further comment.

We sincerely thank the reviewer for the positive evaluation of our revision. We are pleased that the additional functional data and the corresponding revisions have addressed the reviewer's concerns and improved the overall quality of the manuscript.

Reviewer #2 (Remarks to the Author):

Revision Manuscript 555105:

The authors addressed most of the comments successfully, notably by adding the functional data necessary to validate their structural findings. The additions and revisions improved the manuscript substantially.

We sincerely thank the reviewer for the valuable suggestions, which have helped us improve the quality of our manuscript.

However, a few points require clarification or correction:

The authors claim that the clashscore of the structures is less than 10 (see Extended Data Table 1 and rebuttal letter), but the validation reports show substantially different statistics. In particular, two deposited structures have a clashscore of 14. When I requested an updated report for (PDB), it produced a third set of statistics, distinct from both the previous reports and the data table. This raises concerns about consistency between the deposited structures and those used in the structural analysis. Please ensure that all figures, Extended Data Table 1, and structural analyses are updated to match the deposited model.

We sincerely apologize for the inconsistency between the PDB validation report and the statistical summary table. We have updated the statistical summary table and checked it carefully. We greatly appreciate your careful review and insightful comment.

Line 131: Two consecutive dashes appear in the sentence and seem awkward; they could

be removed for smoother reading.

Thank you for your valuable suggestions. We have revised the original text as follows:

(Page 7, line 131)

“Cryo-EM analysis of this sample revealed two distinct LGR4–RSPO2–ZNRF3 complexes with stoichiometric ratios of 1:1:2 and 2:2:2, corresponding to a heterotetramer and a heterohexamer, respectively.”

Line 449: Misspelled Frizzled (not frizzeld; and remove the word “receptors” in the same sentence).

Thank you for pointing this out. We have corrected this mistake as follows:

(Page 32, line 658)

“In addition, the formation of the ternary complex may sterically restrict ZNRF3 from interacting with Frizzled and LRP5/6 for ubiquitination.”

Reviewer #3 (Remarks to the Author):

We would like to thank the reviewer for his/her valuable time in evaluating our manuscript.

Reviewer #4 (Remarks to the Author):

The authors have addressed my concerns through both functional experiments and more cautious statements of the proposed surface interactions in the complexes.

We sincerely thank the reviewer for the positive evaluation of our revision. We are pleased that the additional functional data and the corresponding revisions have addressed the reviewer’s concerns and improved the overall quality of the manuscript.